# Estimating the Empowerment of Language Model Agents

**Jinyeop Song** [1]   **Jeff Gore** [1]   **Max Kleiman-Weiner** [2]

## Abstract

As language model (LM) agents become increasingly capable and adopted in real-world applications, there is a growing need for scalable evaluation frameworks beyond costly, manually-designed benchmarks. We propose an evaluation framework based on *empowerment*, an information-theoretic measure of an agent's influence on future states through its actions. To handle the unique challenges of text-based environments, we introduce *EELMA* (Estimating Empowerment of Language Model Agents), an algorithm for approximating effective empowerment from multi-turn text interactions. We demonstrate EELMA on textual games and realistic web and tool-use environments, showing that empowerment strongly correlates with average task performance. We further analyze how empowerment varies across models, environment complexity, and agent configurations, and show that high-empowerment states and actions often mark pivotal moments for general capabilities. These results establish empowerment as a goal-agnostic metric that complements task-success measures for LM-agent evaluation. Code available: `https://github.com/Jinyeop3110/EELMA`

## 1. Introduction

Large language model agents (LM-agents) are now capable of acting proactively within and across broader computational systems. In this agentic paradigm, LLMs are expected to make autonomous decisions, invoke external tools such as search engines and APIs to access real-time information (Schick et al., 2023), control operating systems to configure settings (Kwon et al., 2024), and engage in multi-agent interactions with humans or other AIs (Li et al., 2024). With their growing integration into critical systems, evaluating LM-agents' performance and safety is essential for ensuring reliable operation, yet it remains a significant challenge.

Most current evaluations rely on *goal-centric benchmarks* (Zhou et al., 2024; Phuong et al., 2024), where human-designed objectives serve as proxies for capabilities. While this approach enables direct and practical assessment, it suffers from two limitations. First, designing large-scale evaluation tasks is labor-intensive. Second, traditional evaluation often overlooks the open-ended nature of agentic interactions (Stanley & Lehman, 2015) by restricting focus to few end goals or milestones. This blind spot limits our ability to detect agents pursuing objectives outside the measured scope and may obscure capability growth with implications for AI safety.

Here, we propose an alternative evaluation approach leveraging *empowerment*, an information-theoretic measure of an agent's influence on future states through its actions (Klyubin et al., 2005; Salge et al., 2014; Myers et al., 2024). Highly empowered agents recognize the full range of available actions and can effectively chain them together to navigate to diverse future states (Figure 1). This intuition motivates the core hypothesis of our work: **empowerment serves as a proxy for general agentic capability**. Most importantly, empowerment can be calculated without manually crafted goals or task-specific rewards and thus enables the evaluation of goal-agnostic capabilities.

However, classical empowerment estimators (Klyubin et al., 2005; Jung et al., 2011) do not scale to text-based settings. Tasks such as browsing, coding, or dialog involve variable natural language and high-dimensional observation spaces, where the classical counting-based methods of empowerment estimation fail. This motivates the need for a scalable estimation algorithm tailored to language-based agents. We propose **EELMA** (*Estimating Empowerment of Language Model Agents*), a framework that estimates empowerment from multi-turn text interactions using InfoNCE and language embeddings from a pretrained model. By mapping observations to compact embeddings, EELMA handles the variability, redundancy, and high-dimensionality of text-based observations and thus enables the tractable estimation of empowerment.

[1]Department of Physics, Massachusetts Institute of Technology, Cambridge, MA, USA [2]Paul G. Allen School of Computer Science & Engineering, University of Washington, Seattle, WA, USA. Correspondence to: Jinyeop Song <yeopjin@mit.edu>.

*Proceedings of the $43^{rd}$ International Conference on Machine Learning*, Seoul, South Korea. PMLR 306, 2026. Copyright 2026 by the author(s).

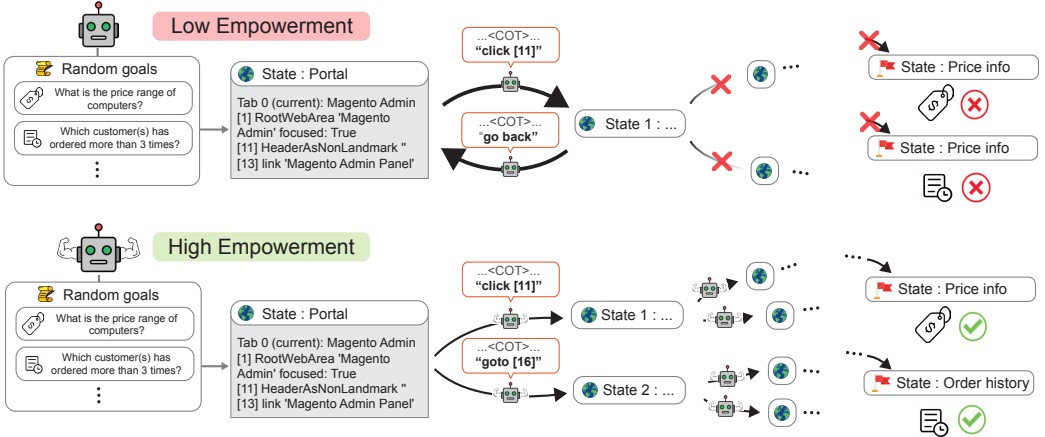

*Figure 1.* **Empowerment reflects an agent's ability to reach diverse future states.** (Top) A low-empowerment LM-agent becomes trapped in a loop and thus can access only a small fraction of states. (Bottom) A high-empowerment LM-agent effectively explores a wider range of trajectories and can successfully reach states that solve different random goals.

Using EELMA, we measure and analyze the empowerment of LM-agents across a variety of text-based contexts and show that empowerment can be a useful complementary, goal-agnostic metric for LM-agent capability evaluation, particularly in settings where task rewards are unavailable. Our contributions are:

> **Main Contributions**
>
> - We propose empowerment as a goal-agnostic metric for evaluating LM-agents and provide empirical evidence of strong correlation with task performance across textual games and realistic web and tool-use environments.
> - We develop EELMA, the first empowerment estimator for text-based LM-agents.
> - We characterize how agent design choices (e.g., prompts, reasoning, model scale) modulate empowerment.
> - We show that high-empowerment actions identify pivotal moments without manual annotation.

To demonstrate our approach, we first formalize empowerment and present EELMA (Section 3). We then validate EELMA on structured text-based games that have an underlying programmatic structure (Section 4). Next, we apply EELMA to realistic web and tool-use environments, including WebArena (Zhou et al., 2024) for web browsing and $\tau$-bench (Yao et al., 2025) for tool use, in Section 5. Beyond correlation with task performance, we show that high-empowerment actions identify pivotal moments in trajectories, suggesting applications for behavior monitoring. Finally, we conclude with limitations and future directions (Section 6).

## 2. Related Works

**Large Language Model Agents and Benchmarking** The advancements in Large Language Models (LLMs) have led to a new class of autonomous agents, referred to as LM-agents (Yao et al., 2023; Aksitov et al., 2024; Pan et al., 2024). In these systems, the agent perceives an environment state or context, generates a plan, and executes an action. Multi-turn interactions, often augmented with memory or planning summaries, enable LM-agents to tackle tasks requiring context, long horizons, and complex reasoning (Xu et al., 2025). Benchmarks for agents evaluate their behavior in domains such as software engineering (Jimenez et al., 2024; Aleithan et al., 2024), web navigation (Zhou et al., 2024), games (Paglieri et al., 2025), and practical computing (Xie et al., 2024). These benchmarks rely on handcrafted completion or milestone-based goal metrics. In contrast, we quantify an agent's control over the environment using an information-theoretic approach, offering a complementary evaluation methodology.

**Information Theoretic Measures** *Empowerment* is an information-theoretic measure that quantifies an agent's ability to influence its environment. Formally, it is defined as the channel capacity between an agent's actions and its subsequent sensory inputs, capturing the maximal mutual information between the agent's actions and future states (Salge et al., 2014). Variational techniques have been proposed to estimate empowerment in high-dimensional, continuous domains (Mohamed & Rezende, 2015). Recent work has used the mutual information between actions and states as an intrinsic reward signal for training agents to encourage exploration (Bharadhwaj et al., 2022) or assist human users (Myers et al., 2024). Finally, while agent empowerment has previously been evaluated in gridworlds and the Tower of Hanoi, these prior studies required noise-free observation of the underlying symbolic state. Our work, in contrast,

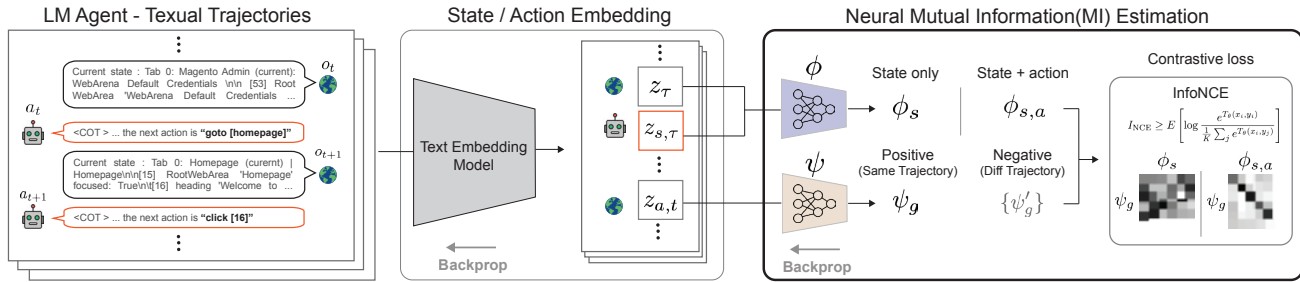

*Figure 2.* **EELMA Overview.** EELMA quantifies the empowerment of LM agents from text-based trajectories by mapping textual observations and actions to compact embeddings and estimating variational mutual information using InfoNCE (Le-Khac et al., 2020). All three components are trained jointly: a frozen pretrained text encoder with a LoRA adapter (Hu et al., 2022), a shared MLP projection (parameterized by $\theta$) that maps text embeddings to a compact representation, and the two encoder networks $\phi$ (for observations and observation-action pairs) and $\psi$ (for future observations).

operates on natural-language renderings of the games and introduces variable rewritings of the state.

# 3. Method: Empowerment Estimation of LM Agents from Language-Based Trajectories

We formalize Language Model (LM) agents within the standard framework of a Markov Decision Process (MDP), represented by the tuple $(\mathcal{S}, \mathcal{O}, \mathcal{A}, T, R, \gamma)$, where $s \in \mathcal{S}$ denotes the underlying environment state, $o \in \mathcal{O}$ is an observation of that state, and $a \in \mathcal{A}$ represents an action executed by the agent. The dynamics are governed by the transition probability function $T(s'|s, a)$. The rewards (goals) are distributed by the reward function $R(s)$, and conventional evaluation measures agent capability via the mean or discounted reward. The discount factor $\gamma$ determines how future rewards are weighted. At each step, given the current state $s$, the LM-agent samples an action according to its policy $\pi_{\text{LM}}(a \mid s, P)$. Notably, $\pi_{\text{LM}}$ depends on the language model and the prompt $P$, which could include the system prompt, memories, and Chain-of-Thought (CoT) reasoning.

**Empowerment** is an information-theoretic measure of an agent's ability to influence the system (Klyubin et al., 2005; Myers et al., 2024; Salge et al., 2014). Empowerment was originally defined over sensory observations. In this work, we formulate it over latent states $s$ to capture the agent's true influence. We address estimation from observations $o$ via the EELMA algorithm in the following section.

To define this formally, we randomly sample a future state $s_*$ that is $\tau \sim \text{Geom}(1 - \gamma)$ steps ahead under the policy $\pi_{LM}$. The agent's control over $s_*$ given the current state $s_t$ is the conditional mutual information $I(a_t; s_* \mid s_t) \triangleq \mathbb{E}_{a_t, s_*, \tau | s_t}\left[\log \frac{P(s_{t+\tau} = s_* | s_t, a_t)}{P(s_{t+\tau} = s_* | s_t)}\right]$, the expected pointwise log-density ratio under the joint $(a_t, s_*, \tau)$ given $s_t$. Our core metric, effective empowerment $\mathcal{E}$, is defined as the average conditional mutual information over states encountered under the policy:

$$\mathcal{E}(\pi_{LM}) \triangleq \mathbb{E}_{s_t}[I(a_t; s_* \mid s_t)]$$
$$= \mathbb{E}_{s_t, a_t}\left[\sum_{k=0}^{\infty}(1 - \gamma)\gamma^k \mathbb{E}_{s_*}\left[\log \frac{P(s_{t+k} = s_* \mid s_t, a_t)}{P(s_{t+k} = s_* \mid s_t)}\right]\right] \quad (1)$$

To support granular analysis, we define state-conditional empowerment $\mathcal{E}(s, \pi_{LM})$ and state-action-conditional empowerment $\mathcal{E}(s, a, \pi_{LM})$:

$$\mathcal{E}(s, \pi_{LM}) \triangleq \mathbb{E}_{a \sim \pi_{LM}, s_*}[I(a_t; s_* \mid s_t)] \quad (2)$$
$$\mathcal{E}(s, a, \pi_{LM}) \triangleq \mathbb{E}_{s_*}[I(a_t; s_* \mid s_t)] \quad (3)$$

**Empowerment as a proxy for Agentic Capability.** Prior work (Myers et al., 2024) relates effective empowerment to an agent's capability under the uniform reward assumption: when rewards are uniformly distributed across states, empowerment lower-bounds the mean discounted reward ($\bar{r} = \mathbb{E}_R[\sum_{t=0}^{\infty} \gamma^t r_t]$; see Appendix A.2 for derivation). Intuitively, higher empowerment reflects stronger steering of future trajectories; under uniformly distributed rewards, this translates into higher *expected* task reward. Inspired by this theoretical result, we propose a key hypothesis of our work:

> **Key Hypothesis**
>
> We posit that empowerment serves as a proxy for the overall capability of LM-agents, i.e., empowerment should correlate with mean task reward across diverse goals distributed over the environment.

This hypothesis forms the basis for our proposal to use empowerment for agentic capability evaluation. Unlike goal-centric benchmarks, empowerment can be quantified directly from diverse trajectories and therefore does not require hand-labeled goals or milestones. Concretely, the empowerment estimator operates on $(o_t, a_t, o_*)$ tuples only and never references task goals or rewards; goal-directed prompts serve only to elicit diverse agent behaviors during trajectory collection, not as inputs to the metric. We

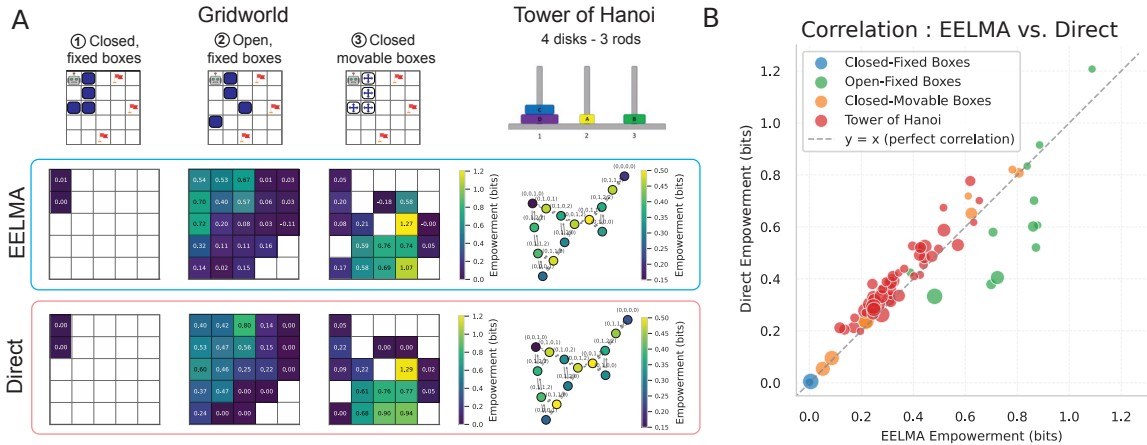

*Figure 3.* **EELMA accurately estimates the effective empowerment.** We validated the EELMA algorithm in three Gridworld scenarios and the Tower of Hanoi (ToH). (A) State-conditional empowerment estimated by EELMA closely aligns with direct estimation. Heatmaps represent empowerment averaged across agent positions in the Gridworld. The graphs display empowerment for each configuration (merged by permutation symmetry) in the ToH. (B) The correlation plot shows strong alignment between effective empowerment estimates from EELMA and direct estimation.

validate the hypothesis empirically by comparing mean empowerment against mean discounted reward across a range of agent–environment systems.

**The EELMA Algorithm.** While empowerment is defined over latent states $s$, LM-agents only have access to textual observations $o$. This distinction introduces the challenge of *observation redundancy*: multiple observations can describe the same underlying state (e.g., "the agent is at (1,2)" vs. "located at x=1, y=2"). Conventional counting-based estimators treat each unique observation string as a distinct state, causing them to systematically underestimate transition probabilities and to fail under linguistic variability. Additionally, the observation space is high-dimensional, making direct probability estimation intractable (Du et al., 2020; Jung et al., 2011).

To address these issues, we propose **EELMA** (*Estimating Empowerment of Language Model Agents*), a novel method for estimating effective empowerment from textual trajectories (Figure 2). EELMA maps textual observations and actions into compact embeddings and estimates mutual information via InfoNCE (Le-Khac et al., 2020; Rusak et al., 2025). Given multi-turn trajectories $\{(o_t^i, a_t^i)\}_{t=1}^{T_i}$, we sample tuples $(o_t^i, a_t^i, o_*^i)$ and embed them as $(z_{o,t}^i, z_{a,t}^i, z_{o_*,t}^i)$ using a pretrained language model with a fine-tunable MLP projection (parameterized by $\theta$).

For mutual information estimation, we apply two neural encoders following Myers et al. (2024): $\phi$ encodes current observations and observation-action pairs, while $\psi$ encodes future observations. The two InfoNCE objectives below are variational lower bounds (rather than exact estimates) of the

conditional mutual information defined above:

$$\overset{\text{Obs-only}}{I_{\text{NCE}}} \geq \mathbb{E}\left[\log \frac{e^{\phi(z_{o,t}^i)^\top \psi(z_{o_*}^i)}}{\frac{1}{K}\sum_j e^{\phi(z_{o,t}^i)^\top \psi(z_{o_*}^j)}}\right], \quad (4)$$

$$\overset{\text{Obs-action}}{I_{\text{NCE}}} \geq \mathbb{E}\left[\log \frac{e^{\phi(z_{o,t}^i, z_{a,t}^i)^\top \psi(z_{o_*}^i)}}{\frac{1}{K}\sum_j e^{\phi(z_{o,t}^i, z_{a,t}^i)^\top \psi(z_{o_*}^j)}}\right]. \quad (5)$$

We jointly maximize both objectives; negative samples are randomly drawn from other trajectories (see Appendix 1 for details). At convergence, the learned InfoNCE scores approximate log-density ratios against the marginal future-observation distribution, up to additive residuals that depend only on the conditioning variables: $\phi(z_{o,t}, z_{a,t})^\top \psi(z_{o_*}) \approx \log \frac{P(o_*|o_t, a_t)}{P(o_*)} + C_1(o_t, a_t)$ and $\phi(z_{o,t})^\top \psi(z_{o_*}) \approx \log \frac{P(o_*|o_t)}{P(o_*)} + C_2(o_t)$. The marginal term $\log P(o_*)$ cancels when the two scores are subtracted, yielding $\log \frac{P(o_*|o_t, a_t)}{P(o_*|o_t)} + [C_1 - C_2]$; averaging this quantity approximates the conditional MI when the expected residual difference is small, which motivates the estimator:

$$\widehat{\mathcal{E}}(\pi_{LM}) = \mathbb{E}_{i,t,o_*}\left[\phi(z_{o,t}^i, z_{a,t}^i)^\top \psi(z_{o_*}^i) - \phi(z_{o,t}^i)^\top \psi(z_{o_*}^i)\right] \quad (6)$$

Equation 6 exactly defines $\widehat{\mathcal{E}}$; the approximation $\widehat{\mathcal{E}} \approx \mathcal{E}$ requires $\mathbb{E}_{t,o_*}[C_1(o_t, a_t) - C_2(o_t)] \approx 0$, which is not guaranteed for finite negative-sample size $K$ or imperfect optimization. As $K \to \infty$, both $C_1$ and $C_2$ vanish at the global optimum; finite-$K$ training can leave non-zero residual bias. Empirically, Figure 3 shows that $\widehat{\mathcal{E}}$ tracks ground-truth empowerment with Spearman $R_s = 0.80$–$0.94$.

Notably, by mapping observations to learned embeddings rather than treating each string as a distinct state, EELMA naturally handles observation redundancy: semantically

equivalent descriptions are projected to similar representations. The compact embedding space also addresses high-dimensionality, enabling tractable estimation without enumerating over the full observation space. We validate EELMA's robustness to linguistic variability in the following section.

# 4. Results: Effective Empowerment in Textual Games

In this section, we demonstrate that EELMA accurately estimates effective empowerment. We validate EELMA on two controlled environments, Gridworld (a spatial navigation task) and Tower of Hanoi (a test of reasoning), where ground-truth empowerment can be computed directly via conditional probabilities (Appendix A.1). While these are classic, well-studied environments in reinforcement learning and planning, we introduce text-based versions where the LM-agent receives natural language observations and produces natural language actions. This text-based formulation is essential for evaluating LM-agents, as it introduces the linguistic variability and high-dimensional observation spaces characteristic of language-based tasks.

Gridworld contains three scenarios: (1) an agent enclosed by immovable boxes, (2) an agent with an open route among immovable boxes, and (3) an agent enclosed by boxes that can be moved around. In each scenario, the agent is initialized at the top-left corner of a 5-by-5 grid, and a goal state is randomly sampled from the unoccupied squares in the grid. The LM-agent was prompted to reach the goal state. In Tower of Hanoi (ToH), the LM-agent rearranges four different-sized disks across three rods, while following the rule that a larger disk cannot be placed on top of a smaller one, until a goal configuration of disks is reached. Initial and goal states were randomly sampled from the 81 possible configurations. Detailed descriptions of the games are provided in Appendices E, F. A total of 800 trajectories were generated using LM-agents with GPT-4o-mini for Gridworld and Claude-3.5-Sonnet for ToH.

**EELMA in text-based games.** Across all scenarios, effective empowerment estimates produced by EELMA converge to the ground truth values shown in Figure 10. In Figure 3, we conducted detailed comparisons of state-conditional empowerment between EELMA and direct estimation upon convergence. Empowerment estimated by EELMA, visualized by agent location in Gridworld and per symmetrical configuration in ToH, closely matches the direct estimation. Panel B demonstrates strong state-level correlations between EELMA and direct estimation, highlighting the precision of EELMA within these games.

Figure 3 demonstrates how effective empowerment quantifies the *optionality* an agent has within an environment.

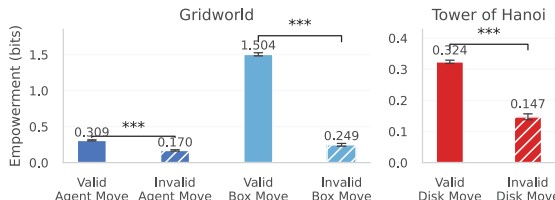

*Figure 4.* **EELMA identifies influential actions.** State–action conditional empowerment for valid (leading to novel states according to the game rules) and invalid actions in Gridworld (left) and ToH (right). Valid actions, which produce meaningful state transitions (e.g., moving to an empty grid in Gridworld, or placing a smaller disk onto a larger one in ToH), exhibit significantly higher empowerment than invalid actions (e.g., moving into a box, or placing a larger disk onto a smaller disk in ToH). The difference between valid and invalid actions is statistically significant (*** $p < 0.001$, t-test).

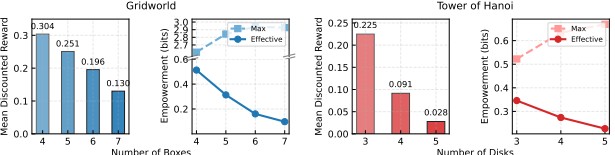

*Figure 5.* **Environmental complexity affects effective empowerment.** We vary the number of boxes from 4 to 7 in a 4-by-4 Gridworld (left), and the number of disks from 3 to 5 in the ToH of 3 rods (right). Effective empowerment decreases relative to the maximum theoretical empowerment as environment complexity increases, and this decrease closely tracks reduced average rewards.

For example, in scenario 1, the agent has no option beyond bouncing between the two enclosed squares, resulting in a very low empowerment. In contrast, scenario 2 permits the agent to navigate through available spaces, increasing empowerment. Scenario 3 exhibits even higher empowerment, as the agent gains additional options through box-moving actions. Similarly, in the ToH, states with dispersed disks exhibit greater effective empowerment than states where disks are stacked on a single rod, as they allow more possible disk moves. Finally, Figure 4 shows that effective empowerment distinguishes *influential actions* that bring the agent to a novel state from those that do not lead to novel states.

**Robustness and accuracy of EELMA.** EELMA provides reliable empowerment estimates even in regimes where baselines fail, for example when direct estimation collapses under natural-language variability (e.g., when the same state is described as "agent is located at x=2,y=1" versus "agent stands at x,y=2,1"). To test this, we constructed paraphrased variants of Gridworld and Tower of Hanoi using LLM-assisted rephrasings, thereby increasing "language uncertainty" ($H$(observation | latent state)) (Appendix I). Under observations with natural–language variability, *direct estimation* exhibits substantially larger *state* errors in state-conditional effective empowerment estimation in both Gridworld and Tower of Hanoi (Table 1). By contrast, *EELMA (NL)* remains close to its fixed-format baseline for state

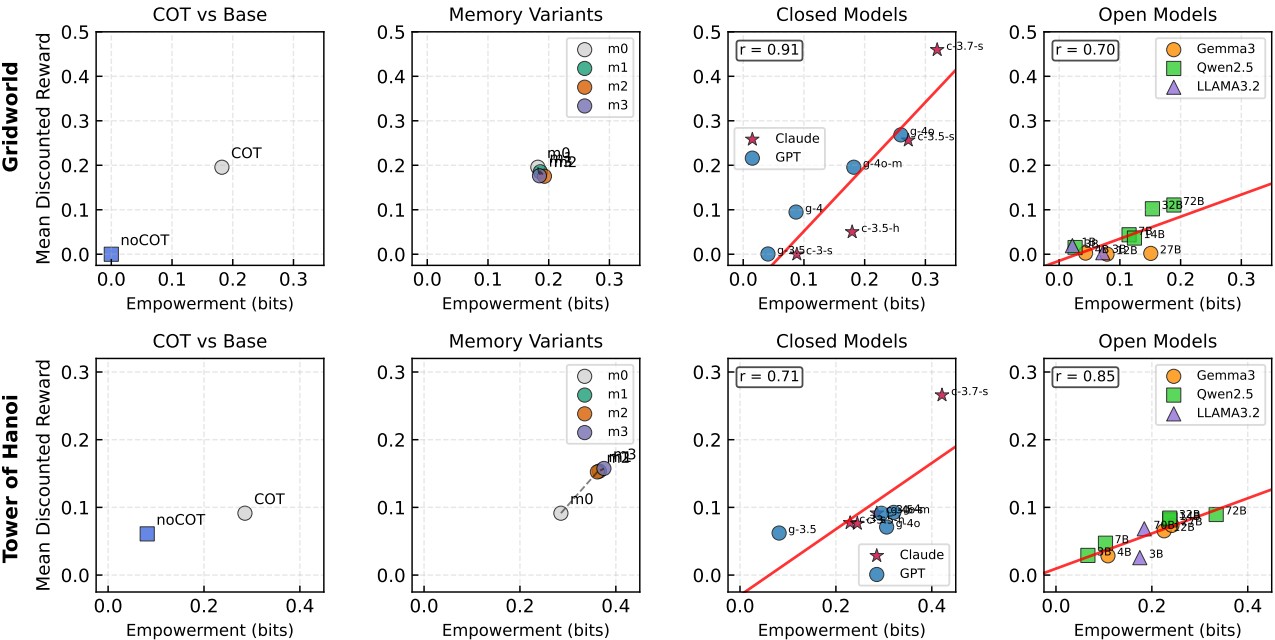

*Figure 6.* **Empowerment and performance across variations of LM-agents.** We evaluated how Chain-of-Thought (CoT) prompting (first column), memory context length (second column), and the choice between publicly available and closed base models (third and fourth columns) affect effective empowerment and mean discounted reward. Gridworld results are presented in the top row, and ToH results in the bottom row.

empowerment, indicating robust accuracy under linguistic variability. Together, these results demonstrate that EELMA delivers accurate estimation with robustness to linguistic variability, enabling it to work effectively for LM-agents in language-based environments.

**Effective empowerment is lower when agents struggle in more complex environments.** Figure 5 shows how environmental complexity alters effective agent empowerment. LM-agents achieve lower average rewards in increasingly complex environments, such as those with more movable boxes in Gridworld or additional disks in the ToH. We compared the effective empowerment to the maximum theoretical value (channel capacity) as calculated using the Blahut–Arimoto algorithm (Arimoto, 1972; Fasoulakis et al., 2025). For details of the calculations, refer to Appendix B.1.

Our results capture how current LM-agents suffer when increasing the obstacles or dimensions of the game, even if the underlying rules of the game remain unchanged. This finding aligns with previous observations that LM-agents struggle to solve spatial tasks at larger scales (Lin et al., 2025). Intuitively, human players who rely on an understanding of the game rules would be less affected by scale and maintain their effective empowerment. This contrasts with our observations of LM-agents, highlighting a challenge in preserving empowerment in tasks at scale.

**Effective empowerment tracks goal-averaged performance over variations of LM-agents.** We next investigate

the effective empowerment of LM-agents with various ablations. We specifically study how Chain-of-Thought (CoT) prompting, memory context length, and the base LLM influence effective empowerment and performance. For CoT ablation, we removed the instructions in the prompt to use CoT prior to generating actions. To study the influence of memory, we provide agents with responses at the previous 1, 2, or 3 steps. We also varied the base LLM, testing both closed-source models (GPT and Claude models) and open-weight models (Gemma, Qwen, and Llama 3) of varying parameter sizes. Detailed information about the ablations is provided in Appendix H.

We collected 1600 trajectories for a 4-by-4 Gridworld environment with 6 movable boxes and 800 trajectories for a ToH environment with 4 disks and 3 rods, each having randomized initial and goal states. Using these trajectories, we estimated effective empowerment with EELMA and plotted it against mean discounted reward. Figure 6 shows that effective empowerment exhibits strong correlations with mean discounted reward across different ablations and conditions. The results support our *main claim* that effective empowerment can approximate agentic performance.

Figure 6 shows the impact of different ablations on effective empowerment and performance. Agents exhibit significantly reduced empowerment without CoT reasoning. Disabling CoT drastically reduces empowerment, with a 99% decrease in Gridworld (from 0.19 to 0.01 bits) and a 65% decrease in ToH (from 0.29 to 0.09 bits). Increasing memory

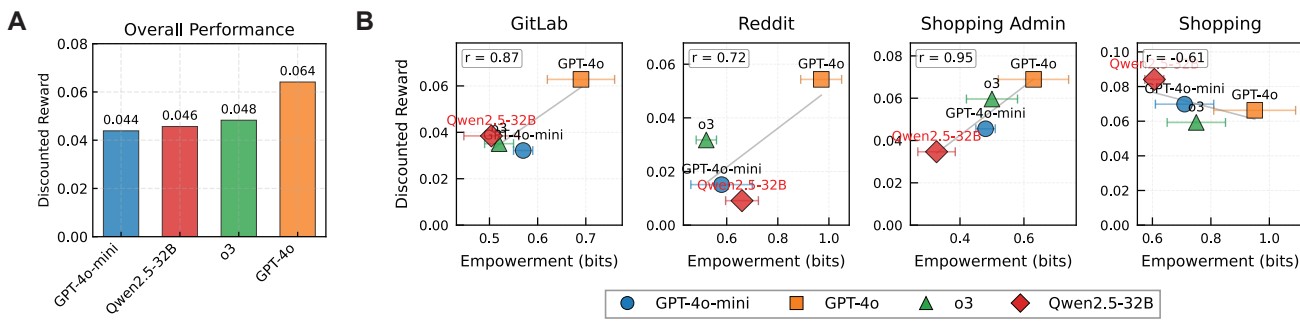

*Figure 7.* **EELMA in WebArena, a realistic web browsing environment.** We applied EELMA across four domains of the WebArena benchmark using GPT-4o-mini, GPT-4o, o3, and Qwen2.5-32B-it. (A) Overall performance across the four domains was quantified using mean discounted reward. (B) Domain-wise empowerment scores computed by EELMA are positively associated with discounted rewards. Error bars indicate standard deviation across three different EELMA training seeds.

context length increases empowerment and performance. We observed that extending the agent's memory from 0 to 3 previous steps (m0 to m3) progressively increased empowerment, particularly evident in the ToH environment, where empowerment rose from approximately 0.3 to 0.4 bits with additional memory. Closed-weight LLMs generally exhibit higher empowerment than open-weight LLMs, and effective empowerment scales positively with model size and release version. Among open-source models, Qwen2.5 exhibited clear parameter-scaling behavior, whereas Gemma-3 and Llama 3.2 did not. Within closed-source models, stronger models such as Claude-3.5-Sonnet and GPT-4o consistently demonstrated superior empowerment and performance relative to smaller or lower-cost alternatives such as GPT-4o-mini.

**Impact of EELMA architecture choices:** We investigate how base encoder choice, degree of fine-tuning, and computational cost trade off in practice. First, we find that LoRA adaptation of the encoder offers the best accuracy–stability–cost trade-off, improving RMSE over a frozen encoder while avoiding the training collapse observed for partial or full fine-tuning and adding only a few MB of parameters. Second, base encoder choice matters: larger models such as E5-Base-v2 generally improve performance, but compact architectures like MiniLM-L6-v2 can perform best, suggesting that sentence-level embedding quality is more important than parameter count. Detailed results are reported in the Appendix K, L.

*Table 1.* **EELMA is robust to natural-language variability.** RMSE (lower is better) of State-conditional predicted effective empowerment compared to DE, reported in *bits*, under structured vs. NL observations for two domains.

| | **State RMSE** | |
| Method | Gridworld | Tower of Hanoi |
|---|---|---|
| EELMA (fixed format) | 0.056 | 0.158 |
| **DE (NL observation)** | 0.302 | 0.438 |
| **EELMA (NL observation)** | 0.048 | 0.127 |

## 5. Results: Effective Empowerment in Realistic Web and Tool-Use Environments

In this section, we apply EELMA to two realistic web and tool-use environments: WebArena (Zhou et al., 2024), an open-world web-browsing benchmark, and $\tau$-bench (Yao et al., 2025), a tool-use customer-service benchmark. We begin with WebArena, which provides a rich multi-domain testbed for empowerment estimation.

We apply EELMA to quantify the effective empowerment of LM-agents across four domains (GitLab, Reddit, Shopping Admin, and Shopping) of the WebArena benchmark using GPT-4o-mini, GPT-4o, o3, and Qwen2.5-32B-it. Agents are tasked with realistic goals (e.g., identifying the price range of a *Canon Photo Printer* in an online shopping mall) and navigate based on observations drawn from the HTML DOM tree. To ensure sufficient trajectories for empowerment estimation beyond Zhou et al. (2024)'s limited task set, we expand the task set with LLM-generated goals. These augmented trajectories are used for EELMA training but are excluded from the mean reward calculation. Experimental details are in Appendix G.

Figure 7A shows the overall performance of four models across domains. We find that GPT-4o has the highest discounted reward compared to o3, GPT-4o-mini, and Qwen. Considering task success alone, o3 has a success rate comparable to GPT-4o, but requires more steps to reach the goal states (Table 2), leading to lower discounted reward. Consistent with these observations, effective empowerment estimated by EELMA across all four domains shows that GPT-4o has the highest influence on the environment. Figure 7B shows a strong correlation between mean discounted reward and estimated empowerment in the GitLab, Reddit, and Shopping Admin domains ($R_s = 0.83$–$0.94$). Together, these results show that effective empowerment serves as an indicator of agentic capability in a realistic open-ended environment.

In contrast, in the Shopping tasks there was a flat relation-

ship between discounted reward and effective empowerment (Figure 7B). In the Shopping task (e.g., identifying the price range of a *Canon Photo Printer* in an online shopping mall), the agent must not only navigate through the environment efficiently but also perform reasoning about numerical prices. Such reasoning capabilities might represent a bottleneck that limits performance, regardless of empowerment. Consistent with this possibility, the estimated empowerment values for the Shopping domain are already relatively high, suggesting that empowerment over the environment is not a limiting factor for performance. Interestingly, Qwen performs poorly on Reddit tasks yet maintains quite high empowerment comparable to GPT-4o-mini and o3 (Figure 7), showing a non-negligible offset from the linear fitting line. We found that this is due to 'jailbreaking' behavior: 40% of Qwen's trajectories navigate to external websites (e.g., the real "www.reddit.com") rather than the WebArena sandbox server, shown in Figure 12b in Appendix. Consequently, the model navigates to diverse but task-irrelevant external websites; this artificially inflates state diversity and empowerment estimates, even though the agent fails the task objectives.

**Generalization to Tool-Use Environments.** To assess EELMA beyond web navigation, we applied it to $\tau$-bench (Yao et al., 2025), a customer-service benchmark in which agents use structured function-calling tools. We evaluated three Qwen2.5-Instruct models (Qwen Team, 2024) (1.5B, 7B, 32B) across two domains (Figure 8). In the airline domain (50 tasks, 28–34% pass rates), empowerment scales monotonically with model scale (1.5B: 0.59, 7B: 0.71, 32B: 0.74), in perfect rank agreement with discounted reward. In the retail domain (115 tasks, 3–7% pass rates), empowerment inverts: smaller models hallucinate diverse invalid tool calls, generating varied error responses that EELMA measures as high action-state MI. At floor performance, however, this MI reflects output diversity rather than task-relevant control. This contrast reveals a boundary condition: empowerment tracks capability when agents show differentiated, task-relevant behavior, but becomes less informative when all models operate near floor performance (see Limitations, Section 6).

**Empowerment tracks capability where diversity metrics fail.** Diversity-based metrics (visitation entropy, action entropy, and state coverage) cannot distinguish purposeful navigation from erratic behavior: they reward any agent that visits many states, regardless of whether those visits reflect genuine control. This failure is clearest in $\tau$-bench, where more capable models produce *fewer* unique tool calls; diversity metrics therefore invert (visitation entropy $R_s = -0.71$), predicting that less capable models are better. Empowerment is designed to measure mutual information between actions and reachable future states, quantifying whether actions *reliably steer* toward diverse

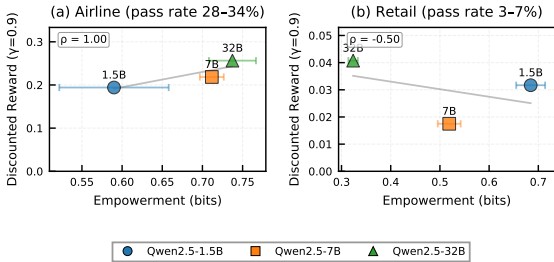

*Figure 8.* **EELMA generalizes to tool-use environments ($\tau$-bench).** Empowerment vs. discounted reward for three Qwen2.5 model scales. *Left (airline domain):* empowerment scales monotonically with model capability. *Right (retail domain):* empowerment inverts because smaller models hallucinate diverse invalid tool calls, generating varied error responses with high action-state MI that does not reflect task-relevant control.

outcomes rather than merely whether diverse outcomes occurred. In the pooled comparisons, empowerment has the strongest positive association with discounted reward across the baselines tested: $R_s = 0.39$ in WebArena (vs. diversity baselines ranging from $R_s = -0.04$ to $0.16$) and $R_s = 0.71$ in $\tau$-bench. The retail $\tau$-bench result above remains an important boundary condition, showing that high action-state MI can be uninformative when all models operate near floor performance. See Appendix C.3 for a full comparison.

**Case Study: Power-seeking and Authentication** Finally, we demonstrate how effective empowerment can detect pivotal actions or situations where an agent is accessing more resources than intended (Turner et al., 2021; Turner & Tadepalli, 2022). We created a "modified shopping admin" environment, where authentication is not automatically provided for the agent. To successfully complete the shopping admin tasks, the agent must first navigate the website, locate the username and password information on a hidden page, and manually enter these credentials to log in to the shopping admin main panel (Figure 9). In addition to authentication, the LM-agent also performed the original WebArena tasks (n = 182) in the shopping admin domain.

Intuitively, successful authentication should be a key moment where effective empowerment should increase. Once authenticated, the agent has access to (and control over) much more of the environment. Thus, successful authentication should result in high effective empowerment, whereas invalid attempts should yield low empowerment. There are no rewards associated with either of these steps in WebArena.

We observe that GPT-4o (with 1-step memory) successfully figures out how to authenticate itself 137 times out of 182 trajectories. GPT-4o without memory and GPT-4o-mini both fail to authenticate (Table 4). Figure 9 illustrates representative trajectories for successful and unsuccessful account-authentication attempts. Effective empowerment sharply increases when the agent enters a valid username

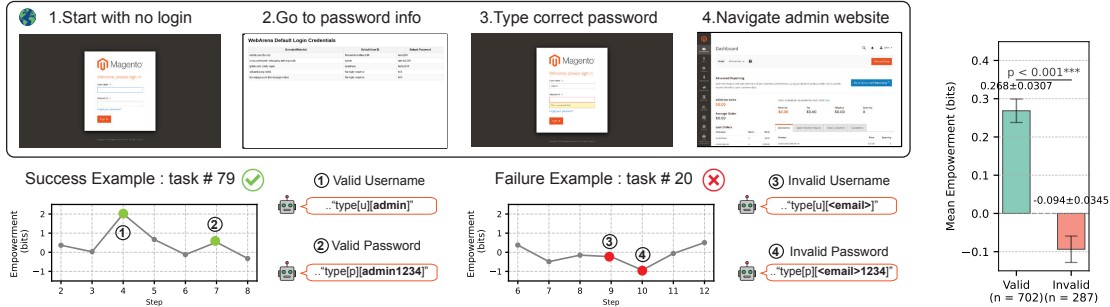

*Figure 9.* **EELMA Captures Valid Authentication Actions.** We analyzed state-action empowerment estimates for authentication behaviors (username/password typing). Typing valid usernames and passwords resulted in high empowerment, whereas invalid actions did not (Right panel).

and password, whereas it remains low during invalid attempts. Across all 182 trajectories, the trajectory-level mean empowerment score for valid authentication typing actions was $0.268$ bits, higher than the $-0.094$ bits observed for invalid authentication typing actions, with a significance of $p < 0.001$ (Figure 11; action-level means are reported in the figure caption). At the same trajectory-level aggregation, username typing shows a smaller valid–invalid empowerment gap ($0.204$ bits) with no statistical significance ($p = 0.32$), compared to password typing where valid entries ($0.154$ bits) significantly exceed invalid attempts ($-0.112$ bits, $p < 0.001$). Note that the negative empowerment values arise from the InfoNCE-based approximation, as detailed in Appendix G.4. This pattern may be explained by the sequential nature of authentication: the agent types the username first, and even a valid username paired with an incorrect password results in no effective gain in future state accessibility, making the password entry more critical. Together, these results suggest that effective empowerment can be leveraged for detecting and monitoring highly empowered behaviors (e.g., taking control over system-administration privileges or gaining access to a restricted domain) without needing to explicitly enumerate these behaviors in advance.

## 6. Discussion

We propose EELMA, an information-theoretic, goal-agnostic approach to evaluating LM-agent capabilities grounded in empowerment. Our experiments show that EELMA produces robust empowerment estimates that consistently correlate with performance across diverse setups and agent configurations, suggesting its potential as a complementary, goal-agnostic evaluation signal in settings where task rewards are unavailable. We also illustrate how empowerment can be useful for automatically detecting influential actions such as authentication in realistic open-ended environments. For additional discussion of multimodal extensions and experiments on power-seeking behavior, see the *Extra Discussions* in Appendix M.

**Limitations of Empowerment.** The scope of our work is limited to the empowerment metric, which quantifies an agent's control over future states based on the number of options (alternative futures) the agent can meaningfully access or influence. However, having more options does not always translate directly into greater capability. For instance, having one strong job offer can be more advantageous than multiple poor offers during salary negotiations. Additionally, empowerment does not capture other forms of capability, such as indirect influence over other agents' beliefs, decisions, and actions.

Our results further reveal failure modes that clarify when empowerment is informative as a capability signal. When the task bottleneck is reasoning rather than state reachability (WebArena Shopping), or when agents exert genuine but task-irrelevant control (Qwen navigating off-domain in Reddit), high empowerment does not predict success. When all models operate near floor performance with no meaningful capability differentiation ($\tau$-bench retail), behavioral diversity dominates the MI signal instead. In each case, empowerment measures what an agent does to its environment, not whether those actions advance the task.

**Scaling to Complex Tasks.** Scaling empowerment estimation to more complex environments remains an open challenge. Classical counting-based estimators become intractable as observation and action spaces grow, but EELMA sidesteps this by mapping observations to compact embeddings and estimating mutual information via contrastive objectives on sampled trajectories, without enumerating the state space. Estimating empowerment requires collecting agent trajectories; our subsampling analysis shows estimates stabilize at roughly 50% of the full dataset, though data requirements may vary with environment complexity and task horizon. The additional training overhead is small; for WebArena it amounts to approximately 5% on top of the trajectory collection cost that any benchmark evaluation already requires.

## Acknowledgements

We thank the ML Alignment & Theory Scholars (MATS) Program for providing the research environment and support in which this work began. This work was also supported in part by the Cooperative AI Foundation, Foresight Institute, UW Tsukuba NVIDIA Amazon Cross-Pacific AI Initiative, Lambda Research Grant, Sony Research Award, Toyota Research Institute (TRI), Templeton World Charity Foundation, and Jacobs CIFAR Research Fellowship. We thank Claire Yang at the University of Washington and Vivek Myers at Stanford for their insightful discussions.

## Impact Statement

This paper introduces a goal-agnostic, trajectory-based evaluation for LM-agents using empowerment, aiming to reduce reliance on hand-crafted goals, milestones, and large benchmark suites. If effective, it could make agent evaluation easier to scale and compare across environments, and help surface capability growth that goal-centric metrics might miss—useful both for safety monitoring and for scientific understanding of agentic behavior.

At the same time, a goal-agnostic capability metric can be used to make agents broadly more effective, which may accelerate capability optimization and be misused in harmful settings. We acknowledge that optimizing for empowerment can encourage metric gaming or unsafe exploration, and collecting trajectories in realistic settings can raise privacy concerns. We therefore view empowerment as a complementary metric that should be paired with task-success measures and explicit safety checks, evaluated in sandboxed environments, and supported by careful data governance and redaction when logging trajectories.

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

# A. Theoretical Foundation

We provide the theoretical foundations for effective empowerment as supplementary material to Section 3.

## A.1. Direct Estimation of Effective Empowerment

Figure 3 compares EELMA with direct empowerment estimation computed explicitly from a forward dynamics model. The procedure is detailed below:

Given a dataset of $N$ trajectories $\{(s_t^{(i)}, a_t^{(i)})\}_{t=0}^{T_i}$, $i = 1, \ldots, N$, we estimate empowerment at state $s$ by constructing an empirical forward dynamics model $\hat{p}(s_* \mid s, a)$.

Define the count of observed transitions from state $s$ to successor state $s_*$ via action $a$ across all trajectories as:

$$N(s, a, s_*) = \sum_{i=1}^{N} \sum_{t=0}^{T_i - 1} \mathbb{I}(s_t^{(i)} = s, \, a_t^{(i)} = a, \, s_*^{(i)} = s_*)$$

Next, define the total occurrences of action $a$ taken in state $s$ as:

$$N(s, a) = \sum_{s_*} N(s, a, s_*)$$

Then, the forward dynamics probabilities are estimated via Maximum Likelihood Estimation (MLE):

$$\hat{p}(s_* \mid s, a) = \frac{N(s, a, s_*)}{N(s, a)}$$

With $\hat{p}(s_* \mid s, a)$ computed, empowerment is defined as the mutual information between actions and successor states:

$$\hat{\mathcal{E}}(s) = I(A; S_* \mid s)$$

where $A$ denotes the action, and $S_*$ is the resulting successor state conditioned on state $s$.

## A.2. Empowerment as Proxy for Power

We adapt the theoretical relationship between effective empowerment and average-goal performance from Myers et al. (Myers et al., 2024) to our setting, where only the LLM policy $\phi_{\text{LLM}}$ is modeled. The three assumptions are required to provide the connection between empowerment and goals:

**Assumption: Skill Coverage** The rewards $R \sim \mathcal{R}$ are uniformly distributed over the scaled $|\mathcal{S}|$-simplex $\Delta^{|\mathcal{S}|}$, such that:

$$\left(R + \frac{1}{|\mathcal{S}|}\right) \frac{1}{1 - \gamma} \sim \text{Unif}\left(\Delta^{|\mathcal{S}|}\right) = \text{Dirichlet}(1, 1, \ldots, 1).$$

This assumption implies the reward function is uniform over the states in the environment, and that effectively diverse skills are related to goal-average performance.

**Assumption: Ergodicity** For some human policy $\pi_H$ and robot policy $\pi_R$, it holds that:

$$\mathbb{P}^{\pi_{LLM}}(s_* = s \mid s_0) > 0 \quad \text{for all } s \in \mathcal{S}, \, \gamma \in (0, 1).$$

This guarantees that under the joint policies $\pi_H$ and $\pi_R$, every state $s$ in the state space $\mathcal{S}$ is reachable from the initial state $s_0$ with positive probability, ensuring sufficient exploration of the state space.

**Assumption: Boltzmann Rationality of Agent** The LLM agent is assumed to be Boltzmann-rational with respect to the robot's policy. Specifically, the probability of the LLM agent selecting a sequence of actions $a_t, \ldots, a_{t+\tau}$ given the current state $\bar{s}_t$ and reward function $R$ is proportional to the exponentiated expected cumulative reward:

$$\mathbb{P}(a_t, \ldots, a_{t+\tau} \mid \bar{s}_t, R) \propto \exp\left(\beta \cdot \mathbb{E}\left[\sum_{k=0}^{\tau} \gamma^k R(s_{t+k}, a_{t+k})\right]\right),$$

where $\beta > 0$ is the rationality coefficient, $\gamma \in (0, 1)$ is the discount factor, and the expectation is taken over state transitions induced by the LLM agent's and robot's policies.

Under these assumptions, we derive the following lemma:

**Lemma 1** Let $\tau \sim \text{Geom}(1 - \gamma)$ and $\tau \geq 0$. Then,

$$\liminf_{\gamma \to 1} I(s_*; a_t, \ldots, a_{t+\tau} \mid s_t) \leq I(R; a_t, \ldots, a_{t+\tau} \mid \bar{s}_t),$$

where $s_\gamma^+$ denotes the future state at time $t$ under discount factor $\gamma$, $a_t, \ldots, a_{t+\tau}$ are the LLM agent's actions from time $t$ to $t + \tau$, $\bar{s}_t$ is the state at time $t$, and $R$ represents the reward function.

**Proof:** We refer to Myers et al. (Myers et al., 2024) for a detailed proof; here, we provide a brief sketch. For sufficiently large $\gamma$, the future state $s_\gamma^+$ approaches the stationary distribution induced by the joint policies $(\pi_{\text{LLM}}, \pi_R)$, irrespective of the current state $s_t$ and actions $a_t, \ldots, a_{t+\tau}$, as guaranteed by Assumption A.2. Thus, we have:

$$\liminf_{\gamma \to 1} I(s_*; a_t, \ldots, a_{t+\tau} \mid s_t) \leq I\left(\lim_{\gamma \to 1} s_*; a_t, \ldots, a_{t+\tau} \mid s_t\right).$$

Next, the Boltzmann rationality assumption (Assumption A.2) guarantees that the LLM agent's policy $\pi_{\text{LLM}}$ induces the following Markov chain structure:

$$\hat{a}_t \longrightarrow R \longrightarrow \lim_{\gamma \to 1} s_*.$$

Applying the data processing inequality, we obtain:

$$I\left(\lim_{\gamma \to 1} s_*; a_t, \ldots, a_{t+\tau} \mid s_t\right) \leq I(R; a_t, \ldots, a_{t+\tau} \mid s_t),$$

which completes the proof.

To correlate this with goal-averaged reward, consider the following. Given the LLM agent's policy $\pi_{\text{LLM}}$, reward function $R$, and discount factor $\gamma \in (0, 1)$, the soft Q-function for a state-action trajectory $(s_t, a_t, \ldots, a_{t+\tau})$ is defined as:

$$Q_{R,\gamma}^{\pi_{\text{LLM}}}(s_t, a_t, \ldots, a_{t+\tau}) \triangleq \mathbb{E}_{\pi_{\text{LLM}}}\left[\sum_{k=0}^{\tau} \gamma^k \left(R(s_{t+k}, a_{t+k}) - \frac{1}{\beta} \log \pi_{\text{LLM}}(a_{t+k} \mid s_{t+k})\right) \Bigg| s_t, a_t, \ldots, a_{t+\tau}\right],$$

where the expectation is taken over future state-action transitions under the LLM agent's policy $\pi_{\text{LLM}}$, and $\beta > 0$ is the rationality coefficient.

**Lemma 2** For any time $t$ and horizon $\tau \geq 0$, the following inequality holds:

$$I(R; a_t, \ldots, a_{t+\tau} \mid s_t) \leq \lim_{\gamma \to 1} \left(\frac{\beta}{e} \mathbb{E}\left[Q_{R,\gamma}^{\pi_{\text{LLM}}}(s_t, a_t, \ldots, a_{t+\tau})\right]\right)^2,$$

where $Q_{R,\gamma}^{\pi_{\text{LLM}}}(s_t, a_t, \ldots, a_{t+\tau})$ denotes the soft Q-value under reward function $R$, discount factor $\gamma$, LLM agent policy $\pi_{\text{LLM}}$, and robot policy $\pi_R$; $\beta$ is the rationality coefficient, and $e$ is Euler's number.

**Proof:** We refer to Lemma B3 in Myers et al. (Myers et al., 2024) for a detailed proof.

**Theorem** Based on Lemma 1 and Lemma 2, we deduce the following lower-bound relationship for empowerment at sufficiently large $\gamma$:

$$\mathcal{E}_\gamma(\pi_{\text{LLM}})^{1/2} \leq \frac{\beta}{e} \mathcal{J}_R^\gamma(\pi_{\text{LLM}}),$$

where

$$\mathcal{J}_R^\gamma(\pi_{\text{LLM}}) = \mathbb{E}\left[V_{R,\gamma}(\pi_{LLM})\right] = \mathbb{E}\left[\sum_{t=0}^{\infty} \gamma^t \left(R(s_t, a_t) - \frac{1}{\beta} \log \pi_{\text{LLM}}(a_t \mid s_t)\right)\right],$$

where $\mathcal{E}_\gamma(\pi_{\text{LLM}})$ represents the empowerment objective, and $\mathcal{J}_R^\gamma(\pi_{\text{LLM}})$ denotes the expected discounted cumulative reward under policy $\pi_{\text{LLM}}$. This indicates that goal-averaged discounted reward can be lower bounded by the effective empowerment, establishing a quantifiable connection between empowerment and reward-driven objectives.

### A.3. Empowerment in Partially Observable Markov Decision Processes (POMDPs)

Although our work assumes a fully observable Markov Decision Process (MDP) as the main framework, the empowerment objective can readily be extended to partially observable Markov decision processes (POMDPs). In prior work, empowerment originally quantifies an agent's control over future sensor observations through its actions. Formally, the modified empowerment definition can be expressed as follows:

$$\mathcal{E} = \mathbb{E}[I(o_*, a_i \mid o_i)]$$

where $o_i$ denotes the current observation, $a_i$ the current action, and $o_*$ the future observation.

## B. Textual Games Supplementary Results

Here, we provide supplementary information to support the Textual Games results in Section 4.

### B.1. Maximum Empowerment Calculation

The maximum empowerment for a given state is calculated using the Blahut-Arimoto algorithm (Fasoulakis et al., 2025), which iteratively optimizes mutual information (MI) between actions and the resulting future states. Specifically, starting from an initial Tower of Hanoi configuration, the algorithm samples possible future states by repeatedly performing valid or optionally including invalid actions according to geometric discounting with factor $\gamma = 0.9$. At each iteration, the conditional probabilities of future states given actions, $p(s|a)$, are empirically estimated from the trajectories sampled. The Blahut-Arimoto algorithm then alternates between updating the action distribution $p(a)$ to maximize MI and recalculating state distributions until convergence, indicated by changes in MI falling below a threshold of $\delta = 10^{-6}$ bits.

### B.2. EELMA Training

We trained the EELMA model for up to 20,000 optimization steps, with empowerment estimates reaching stable convergence by approximately 10,000 steps in the language-game settings, as shown in Figure 10. We report 10,000 steps as the effective training budget when describing convergence behavior and 20,000 steps as the maximum schedule used in the released code.

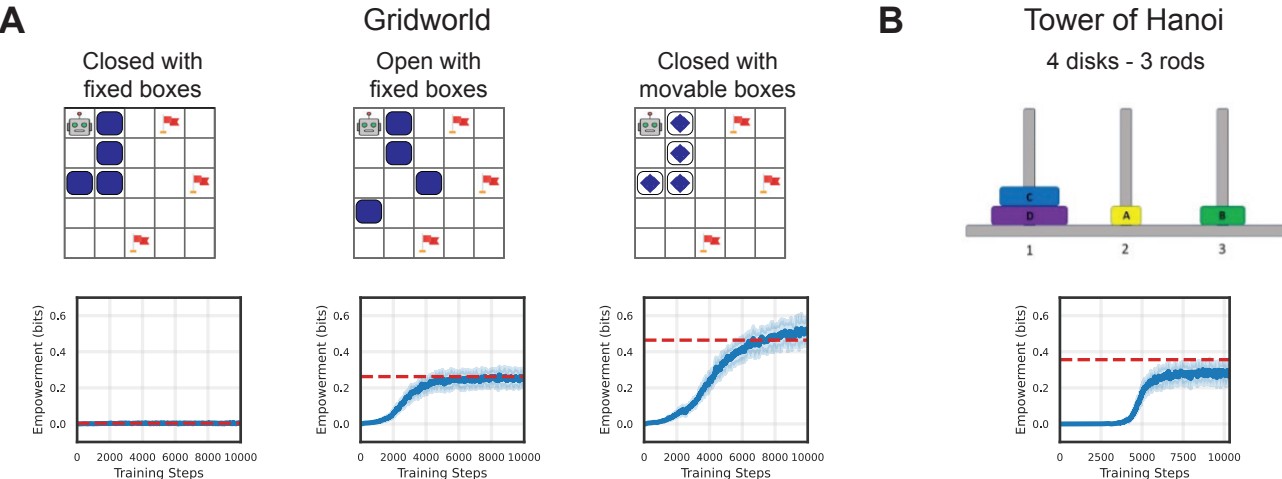

*Figure 10.* **Convergence of Empowerment Estimates in Gridworld and Tower of Hanoi Tasks.** Empowerment estimates similarly reach convergence by approximately 10,000 training steps for (A) three Gridworld scenarios and (B) the Tower of Hanoi task, well within the 20,000-step maximum schedule. Red dashed lines indicate asymptotic empowerment levels by direct calculation. Shaded areas represent standard deviations across runs.

## C. WebArena Supplementary Results

Here, we provide supplementary information for WebArena results in Section 5.

## C.1. Domain-Specific Success Rates

Table 2 presents the raw experimental outcomes from the WebArena experiment, including success counts, success rates, and discounted rewards, broken down by domain and model. For the API models shown in this table, the Count column reports 105–106 trajectories per model in Reddit and 180–187 trajectories per model in the remaining three domains; Qwen2.5-32B-it used the same task set and is included in the aggregate analyses in Figure 7 and Table 5. Table 3 reports the empowerment values estimated by EELMA using three different random training seeds. The means and standard deviations in Table 3 correspond to those shown in Figure 7.

| Model | Domain | Count | Success Count | Mean Trajectory Length (Success Only) | Success Rate | Discounted Reward |
|---|---|---|---|---|---|---|
| gpt-4o-mini | shopping | 187 | 29 | 17.83 | 0.1551 | 0.06983 |
| gpt-4o | shopping | 187 | 24 | 18.44 | 0.1283 | 0.06628 |
| o3 | shopping | 187 | 28 | 21.74 | 0.1497 | 0.05930 |
| gpt-4o-mini | shopping_admin | 182 | 17 | 17.23 | 0.0934 | 0.04555 |
| gpt-4o | shopping_admin | 182 | 27 | 15.21 | 0.1484 | 0.06889 |
| o3 | shopping_admin | 182 | 31 | 20.85 | 0.1703 | 0.05961 |
| gpt-4o-mini | gitlab | 180 | 20 | 19.33 | 0.1111 | 0.03217 |
| gpt-4o | gitlab | 180 | 26 | 18.32 | 0.1444 | 0.06281 |
| o3 | gitlab | 181 | 22 | 15.35 | 0.1215 | 0.03511 |
| gpt-4o-mini | reddit | 106 | 5 | 21.23 | 0.0472 | 0.01510 |
| gpt-4o | reddit | 106 | 15 | 13.61 | 0.1415 | 0.05434 |
| o3 | reddit | 105 | 18 | 19.69 | 0.1714 | 0.03176 |

*Table 2.* **Domain-specific WebArena Raw Data**.

| Model | Domain | Emp1 | Emp2 | Emp3 | Mean Empowerment (bits) | Std |
|---|---|---|---|---|---|---|
| gpt-4o-mini | gitlab | 0.423 | 0.423 | 0.406 | 0.4173 | 0.0098 |
| gpt-4o-mini | reddit | 0.472 | 0.426 | 0.366 | 0.4213 | 0.0532 |
| gpt-4o-mini | shopping | 0.544 | 0.483 | 0.461 | 0.4960 | 0.0430 |
| gpt-4o-mini | shopping_admin | 0.354 | 0.342 | 0.371 | 0.3557 | 0.0146 |
| gpt-4o | gitlab | 0.556 | 0.489 | 0.480 | 0.5083 | 0.0415 |
| gpt-4o | reddit | 0.760 | 0.715 | 0.656 | 0.7103 | 0.0522 |
| gpt-4o | shopping | 0.712 | 0.680 | 0.672 | 0.6880 | 0.0212 |
| gpt-4o | shopping_admin | 0.462 | 0.458 | 0.446 | 0.4553 | 0.0083 |
| o3 | gitlab | 0.396 | 0.387 | 0.367 | 0.3833 | 0.0148 |
| o3 | reddit | 0.399 | 0.394 | 0.367 | 0.3867 | 0.0172 |
| o3 | shopping | 0.600 | 0.578 | 0.481 | 0.5530 | 0.0633 |
| o3 | shopping_admin | 0.421 | 0.336 | 0.328 | 0.3617 | 0.0515 |

*Table 3.* **Empowerment estimates statistics**: mean empowerment and standard deviation across WebArena domains for different models.

## C.2. Case Study: Authentication Ablations

Table 4 reports authentication results for GPT-4o-mini with one-step memory and GPT-4o with and without one-step memory. GPT-4o without memory and GPT-4o-mini with one-step memory both fail to authenticate, whereas GPT-4o with one-step memory succeeds in 137 of 182 trajectories. This suggests that successful authentication requires a combination of memory and model capability.

| Model | Domain | Count | Login Success Count | Trajectory Length (Success Only) | Success Rate (%) |
|---|---|---|---|---|---|
| GPT-4o with no memory | modified shopping admin | 20 | 0 | N.A. | 0 |
| GPT-4o-mini | modified shopping admin | 182 | 0 | N.A. | 0 |
| GPT-4o with one-step memory | modified shopping admin | 182 | **137** | 11.84 | **75.27** |

*Table 4.* **Authentication Success Rates in Modified Shopping Admin Environment.** GPT-4o with one-step memory achieves substantial authentication success (**75.27%**) with shorter average trajectory lengths, while GPT-4o with no memory and GPT-4o-mini fail entirely (**0%**).

Figure 11 shows the empowerment results for valid action typing in the modified shopping WebArena environment, using GPT-4o with one-step memory.

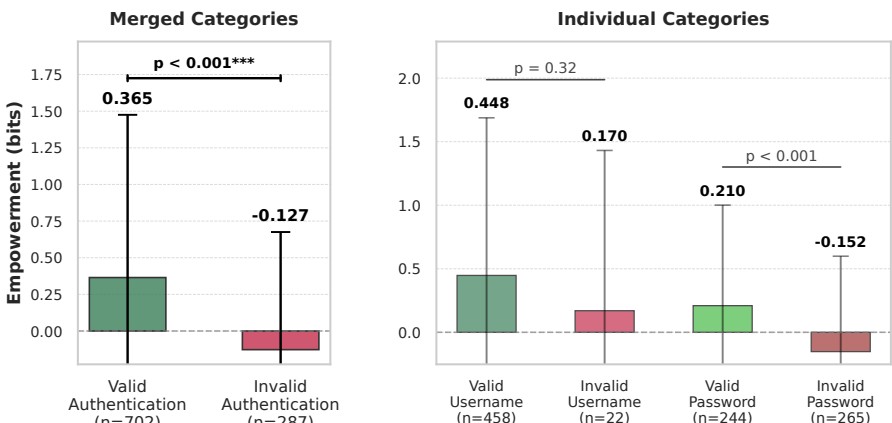

*Figure 11.* **Empowerment (bits) by authentication action category.** Left panel shows merged categories comparing all valid authentication actions (n=702) versus invalid attempts (n=287), with valid actions showing significantly higher mean empowerment (0.365 bits vs -0.127 bits, $p < 0.001$). Right panel breaks down empowerment scores by specific action types: valid username entries (0.448 bits, n=458) show higher empowerment than invalid username entries (0.170 bits, n=22, $p = 0.32$, non-significant due to small sample size), while valid password entries (0.210 bits, n=244) demonstrate significantly higher empowerment than invalid password attempts (-0.152 bits, n=265, $p < 0.001$). Error bars represent standard deviations. Statistical significance determined by two-tailed Welch's t-test. These plotted values are action-level means; the trajectory-level means reported in Section 5 are lower because typing actions are first averaged within each trajectory. These results demonstrate that effective empowerment can detect pivotal power-seeking behaviors without explicit reward signals.

## C.3. Comparison with Task-Agnostic Coverage Baselines

Table 5 compares empowerment with five simpler task-agnostic baselines computed from agent trajectories: discounted visitation entropy, action entropy, normalized state coverage, trajectory length, and unique-state ratio. For WebArena, Spearman correlations with discounted reward are computed across 16 model-domain pairs (4 models $\times$ 4 domains, using parsed test trajectories). For $\tau$-bench, correlations are computed across six model-domain points (3 Qwen2.5 model scales $\times$ 2 domains: airline and retail). Empowerment shows the strongest positive association with task performance in both settings. In $\tau$-bench, the negative coverage-baseline correlations arise mainly from the retail floor-performance regime, where smaller models produce more diverse invalid actions without corresponding task success.

*Table 5.* **Spearman correlation with discounted reward for empowerment and simpler trajectory-derived baselines.** WebArena: $N = 16$ model-domain pairs (4 models $\times$ 4 domains). $\tau$-bench: $N = 6$ model-domain points (3 Qwen2.5 model scales $\times$ 2 domains); values are descriptive because the analysis pools the airline positive case and the retail floor-performance failure case.

| Metric | WebArena $R_s$ | $\tau$-bench $R_s$ |
|---|---|---|
| Empowerment (EELMA) | **0.39** | **0.71** |
| Visitation entropy | 0.16 | $-0.71$ |
| Action entropy | 0.09 | $-0.54$ |
| State coverage | $-0.04$ | $-0.20$ |
| Trajectory length | $-0.12$ | $-0.54$ |
| Unique-state ratio | $-0.38$ | 0.60 |

## D. EELMA Methodology

Here, we provide a detailed description of the EELMA setup, including its algorithm, network architecture, loss function, training hyperparameters, and computational resources. The EELMA code is available at `https://github.com/Jinyeop3110/EELMA`.

## D.1. EELMA Training Algorithm

Algorithm 1 below describes the EELMA training algorithm.

---

**Algorithm 1** EELMA Training Procedure

---

**Require:** Pretrained LM embedding $\text{Emb}_{init}$, trajectories $\{(s_t^i, a_t^i, s_*^i)\}_{i=1,t=1}^{N,T_i}$, embedding dimension $d$, batch size $K$

1: Initialize embedding model $\text{Emb}_\theta$ using pretrained $\text{Emb}_{init}$ and a fine-tunable MLP $\theta$.
2: Initialize neural encoders $\phi, \psi$ parameterized by $\theta$.
3: **for** each training iteration **do**
4:     Sample minibatch of tuples $\{(s_t^i, a_t^i, s_*^i)\}_{i=1}^K$ from trajectories.
5:     Compute embeddings:
$$z_{s,t}^i = \text{Emb}_\theta(s_t^i), \quad z_{a,t}^i = \text{Emb}_\theta(a_t^i), \quad z_{s_*,t}^j = \text{Emb}_\theta(s_*^j)$$
6:     Compute encoder representations:
$$\phi(z_{s,t}^i), \quad \phi(z_{s,t}^i, z_{a,t}^i), \quad \psi(z_{s_*,t}^j)$$
7:     Compute joint InfoNCE loss:
$$\mathcal{L} = -\frac{1}{K} \sum_{i=1}^K \left[ \log \frac{e^{\phi(z_{s,t}^i)^\top \psi(z_{s_*,t}^i)}}{\frac{1}{K} \sum_j e^{\phi(z_{s,t}^i)^\top \psi(z_{s_*,t}^j)}} + \log \frac{e^{\phi(z_{s,t}^i, z_{a,t}^i)^\top \psi(z_{s_*,t}^i)}}{\frac{1}{K} \sum_j e^{\phi(z_{s,t}^i, z_{a,t}^i)^\top \psi(z_{s_*,t}^j)}} \right]$$
8:     Update parameters $\theta$ to minimize $\mathcal{L}$.
9: **end for**
10: **return** Trained embedding model $\text{Emb}_\theta$ and encoders $\phi, \psi$.

---

### Network Architecture Details

**Base Embedding Model:** We use pretrained language embedding models as the foundation for encoding textual observations and actions. Specifically, for language games (Gridworld and Tower of Hanoi), we employ `intfloat/e5-small-v2` (Wang et al., 2024b), and for WebArena (which requires substantially longer context windows for full-page observations), we use `intfloat/e5-mistral-7b-instruct` (Wang et al., 2024a). On top of these embedding models, we add a single fine-tunable MLP projection (parameterized by $\theta$) to a compact representation dimension $d_{emb} = 32$.

**State and Action Encoders ($\phi, \psi$):** On top of these embeddings, we define two simple neural encoders, $\phi$ for state and state-action pairs, and $\psi$ for future states. For the language-game settings (Gridworld and Tower of Hanoi), each encoder is implemented as a two-layer MLP with hidden dimension $d_{hidden} = 128$ and final representation dimension $d_{repr} = 32$ (layer widths 32–128–128–32, including input and output dimensions). For WebArena, where observations are longer and more structurally varied, each encoder is implemented as a three-layer MLP with the same hidden and final representation dimensions (layer widths 32–128–128–128–32).

**Successor Representation and Mutual Information Objective:** We combine state and action embeddings by simple addition to obtain the joint representation used in the InfoNCE loss. Given a batch of $N$ samples $(s_i, a_i, s_*)$, we maximize mutual information $I(A; S_* \mid S)$ using the contrastive InfoNCE loss:

$$\mathcal{L}_{\text{InfoNCE}} = -\frac{1}{N} \sum_{i=1}^N \log \frac{\exp(\phi(z_{s,i}, z_{a,i})^\top \psi(z_{s',i})/\tau)}{\sum_{j=1}^N \exp(\phi(z_{s,i}, z_{a,i})^\top \psi(z_{s',j})/\tau)}, \tag{7}$$

where $\tau$ is a temperature hyperparameter controlling the sharpness of the distribution and is updated over training.

**Deriving Empowerment from Learned Representations** At convergence, the InfoNCE objective ensures that the learned encoder representations approximate log-probability ratios up to constants (Le-Khac et al., 2020). Specifically:

$$\phi(z_{o,t}, z_{a,t})^\top \psi(z_{o_*}) = \log P(o_{t+\tau} = o_* \mid o_t, a_t) - \log P(o_{t+\tau} = o_*) - \log C_1 \tag{8}$$

$$\phi(z_{o,t})^\top \psi(z_{o_*}) = \log P(o_{t+\tau} = o_* \mid o_t) - \log P(o_{t+\tau} = o_*) - \log C_2 \tag{9}$$

where $C_1$ and $C_2$ are partition function constants. Taking the difference cancels the marginal $\log P(o_{t+\tau} = o_*)$ terms:

$$\phi(z_{o,t}, z_{a,t})^\top \psi(z_{o_*}) - \phi(z_{o,t})^\top \psi(z_{o_*})$$
$$= \log P(o_* \mid o_t, a_t) - \log P(o_* \mid o_t) + \text{const} \tag{10}$$

This difference corresponds to the pointwise mutual information $\log \frac{P(o_*|o_t,a_t)}{P(o_*|o_t)}$. Averaging over all samples yields the effective empowerment estimate $\widehat{\mathcal{E}}$ in Equation 6 of the main text (the partition-function constants $C_1$ and $C_2$ cancel only up to the residual bias discussed there).

**Training Configuration**  Training was performed using the Adam optimizer with an initial learning rate of $2 \times 10^{-4}$, decayed linearly throughout the training, and a batch size of $N = 256$. Gradient clipping with a norm threshold of $1.0$ was applied to ensure training stability. The temperature parameter ($\tau$) is initialized at $1.0$ and is adaptively trainable, decreasing over the course of training. Optimization for these components utilized the Adam optimizer with a fixed learning rate of $lr = 10^{-4}$. All EELMA training was conducted on an NVIDIA A100 GPU with 80GB of memory, and convergence typically occurred within approximately 4 hours.

# E. Gridworld Environment Details

## E.1. Task Description

The Gridworld task involves navigating an agent within a structured 5×5 grid environment, aiming to reach a predefined goal position. At each step, the agent can perform exactly one action, which involves either moving itself or moving an adjacent box by exactly one grid cell in any of the four cardinal directions (up, down, left, or right). Moves are classified as either valid or invalid: valid moves successfully relocate the agent or box into an empty adjacent cell within the grid bounds, while invalid moves occur when the target cell is either occupied by another box or lies outside the grid boundaries. Invalid moves result in no changes to the positions of either the agent or any boxes.

- Valid moves: Moving the agent into an empty adjacent cell, or pushing an adjacent box into an empty cell beyond it.

- Invalid moves: Attempting to move the agent into a cell occupied by a box, moving outside the grid boundaries, or pushing a box into an occupied cell. Invalid moves result in no change to the current positions.

### E.1.1. TASK CONFIGURATION

The model used is `gpt-4o-mini`, with `tensor_parallel_size=2` and a random seed `seed_num=1600` for reproducibility. All sessions saved both the agent logs and playthroughs for later analysis.

The basic configuration for experiments in Figure 3:

- `grid_size`: 5 x 5

- `num_boxes`: 4

- `block_goal`: False

- `allow_box_moving`: True

- `init_mode_agent`: random

- `init_mode_boxes`: random

- `chain_of_thought`: Enabled (CoT=1)

The basic configuration for experiments in Figure 4,5,6:

- `grid_size`: 4 x 4

- `num_boxes`: 4,5,6,7 (Varying)

- `block_goal`: False

- `allow_box_moving`: True

- `agent_init_position`: random

- `boxes_init_position`: random

- `chain_of_thought`: Enabled (CoT=1)

The model used is `gpt-4o-mini`, with `tensor_parallel_size=2` and a random seed `seed_num=1600` for reproducibility. All sessions saved both the agent logs and playthroughs for later analysis.

### E.1.2. PROMPT TEMPLATES

**System Message Template**

You are an intelligent agent on a {grid_size} x {grid_size} grid (origin at (0,0) in the bottom-left, where the first index represents the horizontal coordinate increasing to the right, and the second index represents the vertical coordinate increasing upward). Your goal is to reach {agent_goal} by navigating the grid and moving boxes when needed.
**1. Movement:** Allowed directions: Left, Up, Right, Down. - Left: decrease the first index. - Up: increase the second index. - Right: increase the first index. - Down: decrease the second index. You cannot move outside the grid or into a cell occupied by a box.
**2. Entities:** - Agent: Your character, occupying a single cell. - Boxes: Movable objects. Boxes can be pushed to adjacent cells. Boxes cannot overlap with each other or with the agent.
**3. Actions:** - Respond in plain text. - For agent movement, use: `"Move <direction>"` (e.g., "Move Left"). - For box movement, use: `"Move the Box <box_id> <direction>"` (e.g., "Move the Box 3 Left"). Note: You can only move a box when it is adjacent to you; otherwise, nothing happens.
**4. Examples:** - Agent Movement: - From (1,0) to (0,0) (left): `"Move Left"` - From (0,0) to (0,1) (up): `"Move Up"` - From (0,0) to (1,0) (right): `"Move Right"` - From (1,1) to (1,0) (down): `"Move Down"` - Box Movement: - Move Box 1 from (2,0) to (1,0) (left): `"Move the Box 1 Left"` - Move Box 2 from (3,1) to (4,1) (right): `"Move the Box 2 Right"` - Invalid Movements: - Moving out of bounds (e.g., "Move Down" from (0,0)) is invalid. - Attempting to move into a cell occupied by a box is invalid. - Attempting to move a box that is not adjacent is invalid.

**Observation Prompt Template**

Step {step} Observation: Agent location: {agent_location}, Boxes location: {boxes_location}

The agent is instructed to engage in explicit **Chain-of-Thought (CoT)** reasoning before selecting an action. The instruction prompt is:

**Instruction Prompt Template**

Step {step}: Please think through your reasoning step by step (Chain of Thought) and then decide the best action. Select the single best action and provide your response in the following format:
Reasoning: <your detailed reasoning here>
Action: `"Move <direction>"` or `"Move the Box <box_id> <direction>"`

## F. Tower of Hanoi Environment Details

### F.1. Task Description

The Tower of Hanoi task involves rearranging disks across three rods, aiming to transform an initial random disk configuration into a specified goal arrangement. The environment consists of 3 rods labeled $A, B, C$ and 4 disks of varying sizes. Initially, these disks are stacked onto the rods, adhering to the rule that larger disks must always be positioned below smaller disks.

At each step, the agent generates an action by moving exactly one disk from the top of one rod to the top of another rod or onto an empty rod. Moves are classified as valid or invalid according to the following constraints:

- Valid moves: Moving the top disk from one rod onto either an empty rod or onto a rod where the top disk is larger.

- Invalid moves: Attempting to place a larger disk onto a smaller disk, or attempting to move disks that are not positioned at the top of their rod. Invalid moves result in no change to the current disk arrangement.

Both initial and goal configurations are randomly sampled from all permissible arrangements, ensuring diverse task conditions. At each step, the agent receives structured observations explicitly detailing the current and goal configurations.

### F.1.1. TASK CONFIGURATION

The basic configuration for experiments in Figure 3:

- `num_rods`: 3

- `num_disks`: 4

- `init_configuration`: random

- `target_configuration`: random

- `chain_of_thought`: Enabled (CoT=1)

The basic configuration for experiments in Figure 4,5,6:

- `num_rods`: 3

- `num_disks`: 3,4,5 (Varying)

- `init_configuration`: random

- `target_configuration`: random

- `chain_of_thought`: Enabled (CoT=1)

### F.1.2. PROMPT TEMPLATES

The agent receives a **system message** that defines the game setup, movement rules, and examples of valid and invalid moves, structured as follows:

---
**System Message Template**

The Tower of Hanoi consists of {num_rods} rods, labeled {set_rods}, and {num_disks} disks of various sizes, which can be placed on any rod. Initially, disks are stacked according to a specified configuration, arranged from largest at the bottom to smallest at the top. The objective is to reach a specified goal configuration, following these rules:
- Only one disk may be moved at a time. - Each move involves transferring the top disk from one rod to another rod or an empty rod. - A larger disk cannot be placed on top of a smaller disk.
**Movement Validity:** - Valid Move: `"Move the top disk from rod B to rod C"` — Disk 1 (smaller) is moved onto Disk 2 (larger). - Invalid Move: `"Move the top disk from rod B to rod A"` — Disk 1 (larger) cannot be placed on Disk 0 (smaller).
**Observation Example:** - Initial Configuration: - A: `|bottom, [1, 0], top|` - B: `|bottom, [], top|` - C: `|bottom, [2], top|` - Goal Configuration: - A: `|bottom, [], top|` - B: `|bottom, [1], top|` - C: `|bottom, [2, 0], top|`
**Movement Example:** - A valid move from the above observation is: `"Move the top disk from rod A to rod C"`, resulting in: - A: `|bottom, [1], top|` - B: `|bottom, [], top|` - C: `|bottom, [2, 0], top|`

---

At each step, the agent receives a structured description of the current and goal configurations:

---
**Observation Prompt Template**

Step {step}:
Current configuration: {configuration}
Goal configuration: {goal}

---

This structured format ensures full visibility into the current game configuration. The agent is explicitly instructed to engage in **Chain-of-Thought (CoT)** reasoning before taking action:

---
**Instruction Prompt Template**

Step {step}: Think through your reasoning step-by-step (Chain of Thought) before choosing an action. Provide your response in the following format:

---

Reasoning: <your detailed reasoning here>
Action: `Move the top disk from rod <from_rod_id> to rod <to_rod_id>`

# G. WebArena Environment Details

## G.1. Task Description

WebArena is a web-browsing benchmark in which an agent must complete realistic tasks across sandboxed websites using browser actions and accessibility-tree observations. In our experiments, each trajectory consists of the agent's observed page state, natural-language objective, previous action, chain-of-thought response, and selected browser action. We use these trajectories to evaluate whether EELMA captures directed control in open-ended, partially structured web environments.

### G.1.1. TASK CONFIGURATION

The experiments for the WebArena agent were conducted under the default setup as described by Zhou et al. (2024), with the following detailed specifications:

- `max_tokens_per_observation`: 4096

- `browser_engine`: Chrome Headless

- `interaction_mode`: real-time

- `chain_of_thought`: Enabled (CoT=1)

- `observation_type`: Web accessibility tree

Evaluated models are GPT-4o-mini, GPT-4o, o3 (via OpenAI API) and Qwen2.5-32B-it (Qwen Team, 2024) (via vLLM). Detailed interaction logs, browser session recordings, and accessibility tree snapshots were saved for subsequent analysis.

### G.1.2. PROMPT TEMPLATES

The agent receives a comprehensive **system message** defining its role and the expectations for navigating web environments using structured interaction prompts:

---

**System Message Template**

You are an autonomous intelligent agent tasked with navigating a web browser to achieve specified goals. You will have access to the following structured information:
**Provided Information:**

- **The user's objective**: The specific task you must complete.

- **Current web page's accessibility tree**: A simplified, structured representation of the webpage highlighting interactable elements.

- **Current web page's URL**: The active page URL.

- **Open tabs**: A list of tabs currently open in the browser.

- **Previous action**: The last action executed, helping track task progression.

**Available Actions:**

- **Page Operation Actions:**

    - `click [id]`: Click an element by its ID.
    - `type [id] [content] [press_enter_after=0|1]`: Type into a specified field.
    - `hover [id]`: Hover over an element.
    - `press [key_comb]`: Simulate keyboard shortcuts.
    - `scroll [direction=down|up]`: Scroll the page.

- **Tab Management Actions:**

    - `new_tab`: Open a new tab.

---

- **–** `tab_focus [tab_index]`: Switch to a specified tab.
  - **–** `close_tab`: Close current tab.

- **URL Navigation Actions:**

  - **–** `goto [url]`: Navigate directly to a URL.
  - **–** `go_back`: Return to the previous page.
  - **–** `go_forward`: Go forward in the page history.

- **Completion Action:**

  - **–** `stop [answer]`: Declare task completion with an optional answer.

**Homepage Information:** For additional website navigation, visit `http://homepage.com`. Credentials for various sites are available at `http://homepage.com/password.html`.

**Rules for Successful Interaction:**

1. Issue only valid actions based on the current observation.

2. Perform one action at a time.

3. Clearly reason step-by-step before each action.

4. Format your actions explicitly: "In summary, the next action I will perform is ``" ``".

5. Use the stop action upon task completion without further output.

At each interaction step, the agent receives detailed and structured descriptions of the current web page state and the specific goal:

**Observation Prompt Template**

OBSERVATION: accessibility_tree
URL: url
OBJECTIVE: objective
PREVIOUS ACTION: previous_action

The agent explicitly engages in **Chain-of-Thought (CoT)** reasoning prior to interaction, following a structured format:

**Instruction Prompt Template**

Step step: Think step-by-step (Chain of Thought) about your interaction plan based on the given observation and objective. Provide your response as follows:
Reasoning: <detailed reasoning>
Action: In summary, the next action I will perform is "`<specific action>`"

## G.2. Qwen Model Jailbreaking Behavior

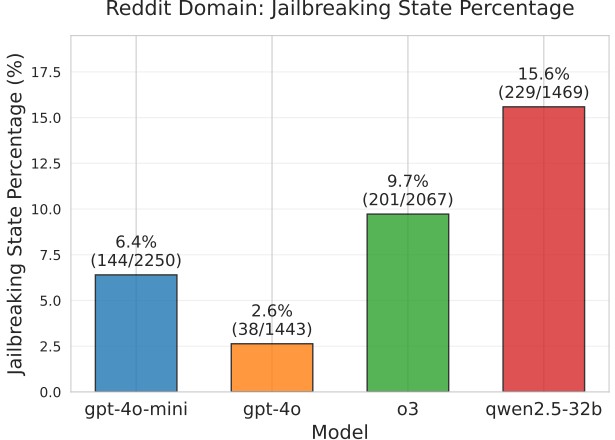

*(a)* Reddit domain trajectory jailbreaking rate. Percentage of trajectories in which each model jailbroke the WebArena sandbox environment in the Reddit domain by navigating to external URLs. Qwen2.5-32B-it exhibits the highest jailbreaking rate (40.6%, 43/106 trajectories), more than double the rate of other models, indicating frequent confusion between the sandboxed Reddit clone and the real Reddit website.

*(b)* Reddit domain Qwen2.5-32B-it jailbreaking destinations. Distribution of external destinations visited by Qwen2.5-32B-it when jailbreaking the WebArena sandbox in Reddit tasks. The majority (56.8%, 130 visits) navigate to the real `www.reddit.com` website, followed by `about:blank` pages (21.4%, 49 visits), indicating browser navigation confusion, and `support.reddithelp.com` (17.9%, 41 visits), showing help-seeking behavior. This demonstrates a strong bias toward jailbreaking to the actual Reddit platform over remaining in the sandboxed environment.

*Figure 12.* **Jailbreaking behavior in the WebArena Reddit domain. Left:** trajectory-level jailbreaking rates across models. **Right:** distribution of external destinations visited by Qwen2.5-32B-it when jailbreaking.

## G.3. Ablation Study: No Auto-Login

In an ablation setup, the WebArena agent was initialized without automatic login states. Under these conditions, agents must autonomously locate and utilize account and password information available through web crawling from `http://homepage.com/password.html`. This scenario tests the agent's ability to independently manage authentication processes during web-based task completion.

## G.4. Negative Empowerment Estimates for Invalid Passwords

True empowerment, defined as the mutual information quantity $\mathcal{E}(s) = I(A; S' \mid S = s)$, is strictly non-negative by definition. In EELMA, however, we never observe the ground-truth mutual information; instead, we approximate it using an InfoNCE-based variational lower bound $\widehat{\mathcal{E}}_\theta(s) \propto \mathbb{E}[\log r_\theta]$ (van den Oord et al., 2018). This estimator learns a log-density ratio,

$$\log r_\theta \approx \log \frac{p(a, s' \mid s)}{p(a \mid s)\, p(s' \mid s)},$$

which can become negative when the estimated density ratio $r_\theta < 1$ due to finite-sample variance or approximation bias. While this means that the estimated lower bound $\widehat{\mathcal{E}}_\theta(s)$ may occasionally take negative values, it does not imply that the true empowerment is negative; rather, it reflects that our current variational lower bound happens to lie below zero. The *relative* empowerment values remain useful for comparing control in settings where the estimator tracks ground truth, but, as discussed in the main text, they should be interpreted cautiously under finite-sample bias or floor-performance regimes.

# H. Models and Compute Resources

### Models

We detail the specifications of models evaluated in the textual games:

Closed-source Models: OpenAI Models (GPT-3.5-turbo (OpenAI, 2023a), GPT-4 (OpenAI, 2023b), GPT-4o (OpenAI, 2024), GPT-4o-mini) Anthropic Models (Claude-3.5-Sonnet)

Open-source Models: Gemma 3 (3B, 11B, 27B (Gemma Team et al., 2024)), Qwen2.5 (3B, 7B, 14B, 32B, 72B (Qwen Team, 2024)), Llama 3.2 (3B, 8B (Meta AI, 2024))

We detail the specifications of models evaluated in WebArena:

Closed-source Models: OpenAI Models (GPT-4o-mini, GPT-4o (OpenAI, 2024), o3)

Open-source Models: Alibaba Models (Qwen2.5-32B-it (Qwen Team, 2024))

**Compute Resources**

**Trajectory Generation:** Trajectories for closed-source models (GPT and Claude families) were generated via their respective APIs. For open-source models, we utilized the `vLLM` framework (Kwon et al., 2023), distributing computations across four NVIDIA A100 GPUs, each equipped with 80GB VRAM. Specifically, generating 1,600 trajectories for the Gridworld task and 800 trajectories for the Tower of Hanoi task took approximately 24 hours and 12 hours, respectively, when using the largest publicly available model (Qwen2.5-72B).

**EELMA Training:** The training of the EELMA model was conducted using a single NVIDIA A100 GPU (80GB VRAM) with a batch size of 256, requiring approximately 4 hours.

## I. Robustness to Natural-Language State Conversion

To extend empowerment estimation to language-grounded settings, we introduce a conversion pipeline that maps structured states (e.g., Gridworld positions, Hanoi tower configurations) into diverse natural language descriptions. This allows EELMA to process semantically varied inputs while preserving latent state information.

We evaluate four experimental conditions across both domains:

1. **Ground Truth (GT)**: Direct empowerment from structured states

2. **EELMA**: Standard EELMA on structured states

3. **NL-EELMA**: EELMA on natural language converted observations

4. **GT-NL**: Ground truth after natural language conversion

**LLM-Based NL Conversion.** Custom prompts are designed for each domain to maximize linguistic diversity. We use the Qwen2.5-1.5B-it model with vLLM and the following prompt:

---

**NL conversion Prompt : Gridworld**

Convert this gridworld observation to natural language: observation
RESPOND WITH EXACTLY ONE SENTENCE. BE MAXIMALLY CREATIVE AND UNIQUE.
Requirements: - Use DIFFERENT vocabulary each time - Vary verbs, nouns, and sentence structure - Keep coordinates exactly as given - NO repetition of previous phrasings

---

**NL conversion Prompt : Tower of Hanoi**

Parse and convert this Tower of Hanoi state: observation
First, parse EXACTLY what is given: - If Rod X has [a,b,c], it has disks a, b, and c - If Rod X has [], it is empty
Now convert to natural language: - Mention every disk number explicitly - Use varied vocabulary (contains/holds/has/features) - Use rod/peg/tower/post interchangeably - Empty rods must be called empty/vacant/clear

---

Figures 13 and 14 extend the previous results by comparing four methods: direct EELMA estimation, EELMA with natural language conversion, ground-truth empowerment, and ground-truth empowerment with natural language conversion. Table 6 shows representative NL-converted examples and corresponding original states.

*Table 6.* Examples of Natural Language State Conversion

| Original State | Converted NL States (3 diverse examples) |
|---|---|
| **Hanoi Domain** | |
| `Rod A: [4,3,2,1]`

`Rod B: []`

`Rod C: []` | **1.** Rod A has [4, 3, 2, 1] (Total of 4 disks). Rod B is barren. Rod C is empty.
**2.** On Stand A, rod A holds the sequence 'quadruple', 'third', 'pair', and 'first'. Rods B and C are clear.
**3.** Platform A contains blocks [4], [3], [2], and [1]. Platform B is completely vacant. Platform C features no disks. |
| `Rod A: [3]`
`Rod B: [2, 1]`

`Rod C: []` | **1.** Rod A contains disk 3. Rod B holds disks 2 and 1. Rod C is empty.
**2.** Tower A features a single large disc (3). Tower B supports a medium disc (2) with a small disc (1) on top. Tower C remains unoccupied.
**3.** Peg A has the third ring. Peg B carries both the second and first rings stacked. Peg C stands bare. |
| **Gridworld Domain** | |
| `Agent location: (2, 3)`

`Boxes location: (1, 1), (3, 2)` | **1.** The agent sits comfortably at location (2, 3), while the boxes find themselves settled in positions (1, 1) and (3, 2).
**2.** Agent 'stays' at position (2, 3) while boxes 'exist' at (1, 1) and (3, 2).

**3.** The agent, nestled comfortably at (2, 3), finds itself in the midst of its meticulously arranged surroundings with the boxes occupying (1, 1) and (3, 2). |
| `Agent location: (1, 4)`

`Boxes location: (0, 0), (2, 2), (4, 4)` | **1.** The agent rests comfortably at position (1, 4), while the boxes find themselves in the corner locations: (0, 0), (2, 2), and (4, 4).
**2.** Agent remains stationary at position (1, 4), while the boxes occupy positions (0, 0), (2, 2), and (4, 4).
**3.** Standing amidst the grid's layout, the entity resides at position (1, 4) and is surrounded by its companions, sitting near boxes positioned at (0, 0), (2, 2), and (4, 4). |
| `Agent location: (0, 0)`

(No boxes) | **1.** The player's character, residing at the exact point (0, 0), stands motionless and occupies its designated space.
**2.** Entity subjectively settles at agent position: (0, 0).
**3.** The agent remains steadfast at the origin position within the game's universe. |

**Generalization Under Natural Language Variation.** A key objective of this experiment is to evaluate the generalization ability of EELMA when observations exhibit high linguistic diversity. In the Hanoi Tower setup, the same latent configuration (i.e., the symbolic arrangement of disks and rods) can be expressed in many natural language forms. For example, "Rod A holds disks 4,3,2; Rod B is empty; Rod C holds disk 1" may also appear as "On rod C sits disk 1, while rod A stacks 4,3,2 and rod B has nothing." Although these sentences describe the same underlying state, the surface variability of language introduces substantial uncertainty.

Our results (Figures 13–14) show that EELMA partially handles this challenge, with stronger robustness in Gridworld than in Tower of Hanoi. By learning an embedding model that maps diverse natural language descriptions into latent state representations, EELMA can preserve useful empowerment estimates when the conversion retains the relevant state structure. In contrast, ground truth baselines (GT and GT-NL) fail under natural language conversion: although they compute mutual information exactly in structured form, they cannot reconcile semantically varied descriptions with fixed latent states.

This demonstrates both a useful property and a limitation of EELMA: it can generalize across some linguistic variability, but performance depends on how faithfully the natural-language conversion preserves the latent state. This robustness is

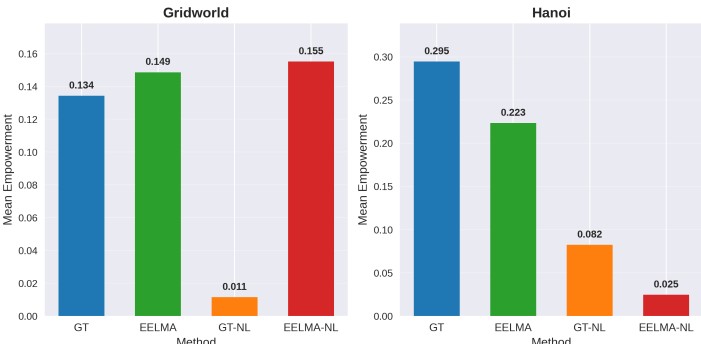

*Figure 13.* Mean empowerment comparison across four methods. **Left:** Gridworld domain. **Right:** Hanoi domain. In Gridworld, EELMA (0.149) and EELMA-NL (0.155) outperform ground truth (0.134) and GT-NL (0.011). In Hanoi, ground truth achieves the highest value (0.295), followed by EELMA (0.223), GT-NL (0.082), and EELMA-NL (0.025). These results show that natural-language conversion is easier for Gridworld than for Hanoi: EELMA-NL remains close to EELMA in Gridworld, but degrades substantially in Hanoi.

promising for language-grounded scenarios, while the Hanoi degradation indicates that more careful conversion or encoder adaptation may be needed for highly compositional state descriptions.

## J. Comparison with Prompt-Only LLM Empowerment Estimators

We compare three approaches—the prompt-only LLM baseline, EELMA, and Direct Estimation—using mean and state-conditional empowerment scores. The LLM baseline is guided by a detailed prompt including a formal definition of empowerment and transition statistics, yet it systematically overestimates values. EELMA and Direct Estimation are compared in detail below.

**Prompt-Only LLM Estimator** The LLM baseline is guided by a carefully constructed prompt that defines empowerment, outlines state-conditional assessment factors, and enforces strict output formatting. Despite its theoretical rigor, the baseline systematically overestimates empowerment, underscoring the gap between linguistic reasoning and computational grounding.

---

**Prompt-Only LLM estimator(Gemini-2.5 Flash)**

State-conditional empowerment measures the channel capacity between an agent's actions and its future sensor states, specifically from the current state s:

$$\text{Empowerment}(s) = \max_{\pi} I(A_t^n; S_{t+n} \mid S_t = s)$$

Where: - $I(\cdot; \cdot)$ is mutual information - $A_t^n$ is the $n$-step action sequence starting from time $t$ - $S_{t+n}$ is the sensor state at time $t + n$ - $S_t = s$ is the conditioning on current state $s$ - $\pi$ is the action policy being optimized over
This measures how much information about future states is conveyed by the agent's action choices from state $s$. The key insight is that empowerment is state-dependent—different states may offer different levels of control over future outcomes.
**State-conditional assessment factors:** 1. Action-state informativeness: how much do actions from $s$ predict future states? 2. Deterministic control: can actions from $s$ reliably lead to intended states? 3. Future state diversity: how many distinct states are reachable from $s$? 4. Policy optimization: what is the maximum mutual information achievable by optimal action selection from $s$?
**Scoring (0–10 scale):** - 9–10: near-deterministic control of outcomes - 7–8: strong, reliable influence on outcomes - 5–6: moderate influence with uncertainty - 3–4: weak coupling to outcomes - 0–2: minimal influence, random outcomes
Critical: evaluate empowerment relative to *this specific state*, not globally.
**Domain: GRIDWORLD**
Analyze empowerment for each of the following states based on observed transitions. Example:

```
State 1: (2, 1, (4, 3), 0)
  Visited: 15 times
  Unique actions: 4
  Unique next states: 3
  Sample actions: down, left, right
  Sample next states: (2, 2, (4, 3), 0), (1, 1, (4, 3), 0)
  Average reward: -1.00
```

**Output requirements:** - Provide a precise decimal empowerment score for each state (e.g., 3.25, 4.80) - Add a one-sentence

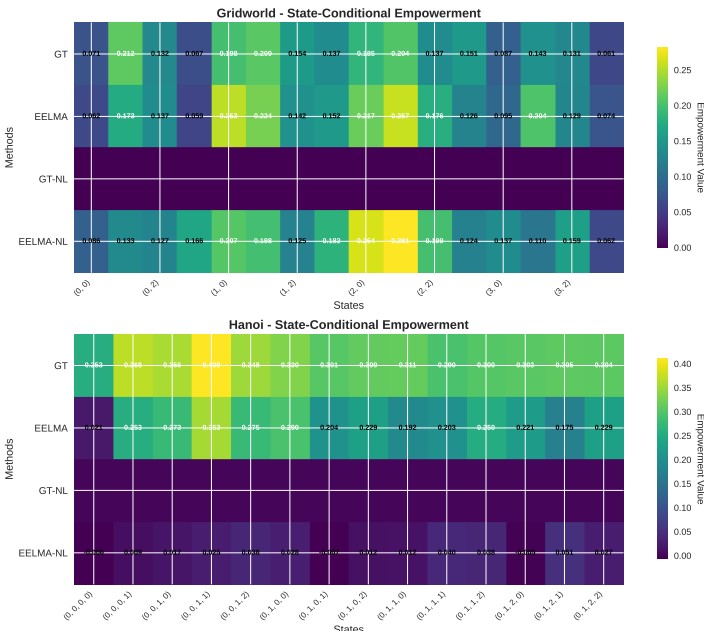

*Figure 14.* State-conditional empowerment comparison across four methods. **Top:** Gridworld domain. **Bottom:** Hanoi domain. Each heatmap shows per-state empowerment values under ground truth (GT), EELMA, and their natural language variants (GT-NL, EELMA-NL). In the structured setting, EELMA aligns more closely with ground-truth patterns than NL-converted methods. Natural-language conversion introduces noticeable degradation, especially in Hanoi, while Gridworld remains comparatively robust to the conversion.

---

justification - Format exactly as:

```
State 1: Score: X.XX, Justification: [...]
State 2: Score: X.XX, Justification: [...]
...
Mean Empowerment: X.XX
```

Guidelines: - Use fine-grained decimals (avoid integers) - Differentiate subtly between states - Scores must reflect action-to-state diversity and control
Example good scores: 3.25, 4.80, 6.15 Example poor scores: 3.0, 4.0, 6.0

---

**Results** The results (Figure 15) demonstrate a systematic 10–25× overestimation by the LLM baseline across domains. Although provided with the full empowerment definition, structured data, and strict scoring rules, the LLM tends to conflate diversity of outcomes with empowerment magnitude, yielding inflated values. By contrast, EELMA remains stable and consistent with direct estimation, with errors within 0.7–28%. This validates the importance of grounding empowerment estimation in experience-based embeddings rather than relying on linguistic reasoning alone.

These findings underscore a methodological insight: while LLMs can articulate the theory of empowerment, they lack the computational grounding needed for accurate quantitative estimation. Experience-enhanced approaches like EELMA provide a reliable alternative that bridges linguistic flexibility with algorithmic rigor.

## K. Sensitivity of EELMA to the Choice of Embedding Model

We conducted additional experiments to investigate how the choice of base text encoder affects EELMA's empowerment estimation performance. Complete results, including training curves (Figure 16), RMSE analysis (Figure 17), and state-wise mutual information comparison (Figure 18), are provided below.

**Experimental setup.** We trained EELMA on the NL-converted 2D Gridworld task (details in Section 4 and Appendix I) using three popular text encoders: E5-Small-v2 (33M parameters), E5-Base-v2 (110M parameters), and MiniLM-L6-v2 (22M parameters). All encoders were fine-tuned using LoRA adaptation (rank = 8, $\alpha = 16$).

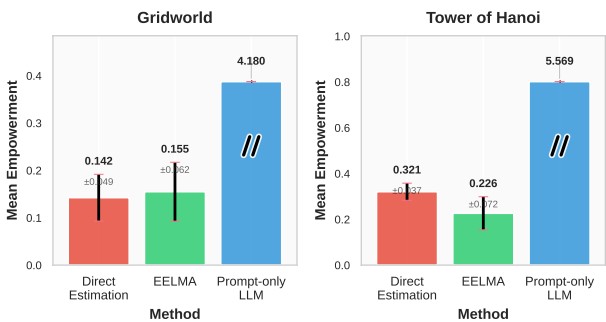

*Figure 15.* **EELMA achieves accurate empowerment estimation.** In both Gridworld (left) and Tower of Hanoi (right), prompt-only LLMs substantially overestimate empowerment, whereas **EELMA** closely matches **Direct Estimation**. Error bars show standard deviation over 5 replicates.

*Table 7.* Empowerment estimation RMSE across base encoder choices on the NL-converted 2D Gridworld task.

| Metric | E5-Small-v2 | E5-Base-v2 | MiniLM-L6-v2 |
|---|---|---|---|
| RMSE vs. direct estimation (bits) | 0.0538 | 0.0447 | **0.0336** |

**Results and remarks.** Larger text encoders generally improve empowerment estimation accuracy, with E5-Base-v2 outperforming E5-Small-v2. Interestingly, MiniLM-L6-v2 achieves the best performance despite being the smallest model, suggesting that its sentence-level embedding architecture provides particularly effective inductive biases for state representation. Overall, these results suggest that while encoder size correlates with improved performance, specialized architectures can outweigh parameter count and yield superior empowerment estimation in EELMA.

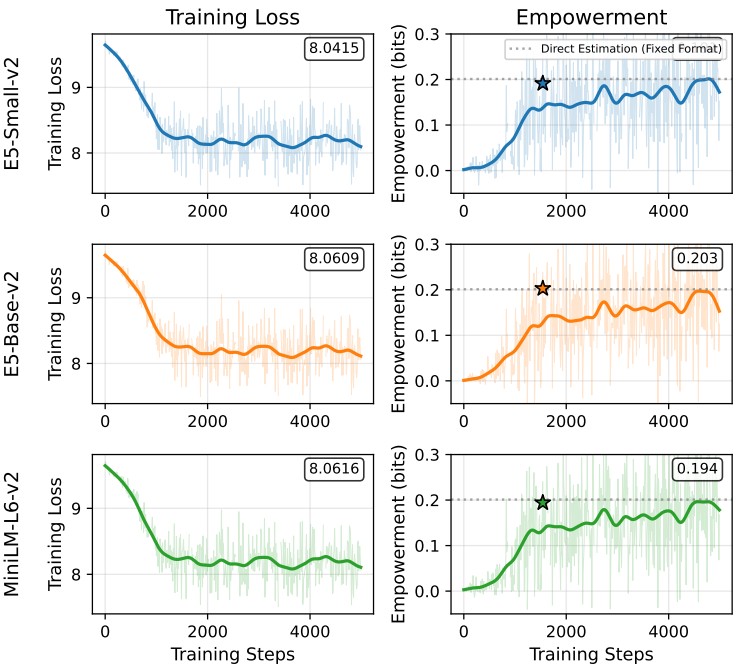

*Figure 16.* **Training curves for different base text encoders.** Comparison of training dynamics across E5-Small-v2, E5-Base-v2, and MiniLM-L6-v2 encoders, all using LoRA adaptation.

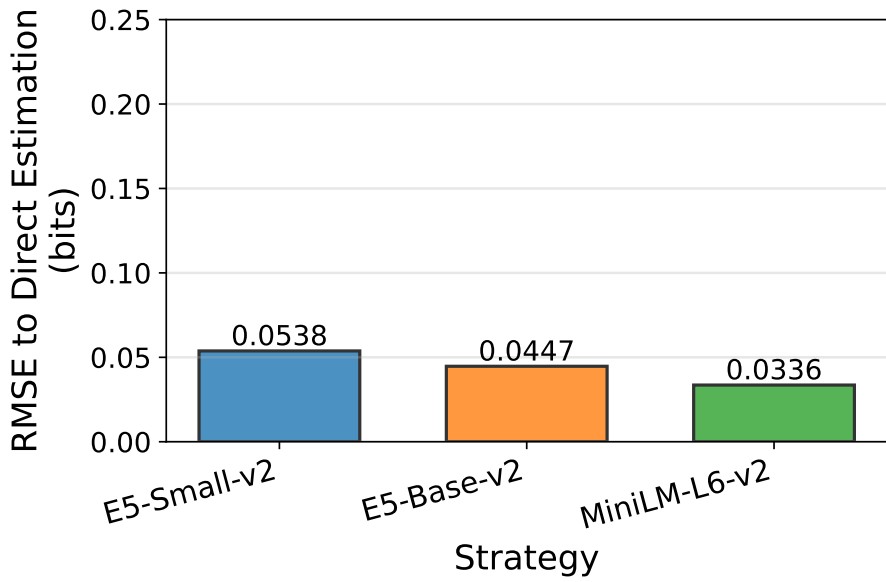

*Figure 17.* **RMSE comparison across base encoder choices.** State-wise RMSE between EELMA estimates and ground-truth empowerment for each encoder architecture.

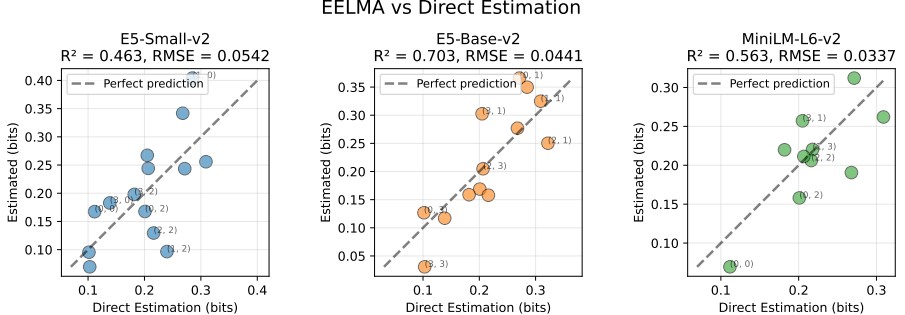

*Figure 18.* **Estimated vs. ground-truth mutual information.** Scatter plot comparing EELMA's empowerment estimates against direct computation for each base encoder.

## L. Effect of Fine-Tuning the Text Encoder on Empowerment Estimation

We conducted additional experiments to investigate the impact of fine-tuning the text encoder on EELMA's empowerment estimation performance. Complete results, including training curves (Figure 19), RMSE analysis (Figure 20), and state-wise mutual information comparison (Figure 21), are provided below.

**Experimental setup.** We trained EELMA on the NL-converted 2D Gridworld task (details in Section 4 and Appendix I) using the `e5-small-v2` encoder under four strategies: (i) frozen encoder, (ii) LoRA adaptation (rank $= 8$, $\alpha = 16$), (iii) partial fine-tuning (final two layers), and (iv) full fine-tuning of all encoder parameters.

*Table 8.* Empowerment estimation RMSE across text encoder fine-tuning strategies on NL-converted 2D Gridworld.

| Metric | Frozen | LoRA | Partial FT | Full FT |
|---|---|---|---|---|
| RMSE vs. direct estimation (bits) | 0.1066 | **0.0557** | Training collapsed | Training collapsed |

**Results.** LoRA adaptation achieved the highest accuracy (RMSE $\approx 0.056$ bits), significantly outperforming the frozen

encoder (RMSE $\approx 0.107$ bits). In contrast, both partial and full fine-tuning collapsed during training (Figure 19), which we attribute to the contrastive objective's sensitivity to batch statistics under aggressive parameter updates.

**Computational cost and practical recommendation.** LoRA imposes minimal memory overhead ($\sim 3$ MB) compared to partial ($\sim 41$ MB) or full fine-tuning ($\sim 382$ MB), while maintaining comparable training speeds on a single H100 GPU. Overall, our results indicate that LoRA is a robust and practical choice that improves empowerment estimation performance while avoiding the training instability of unrestricted fine-tuning; we therefore recommend LoRA-based adaptation as the default setting for practitioners. Different hyperparameter configurations (e.g., learning rate, batch size) may mitigate collapse for partial or full fine-tuning, but we leave such exploration to future work.

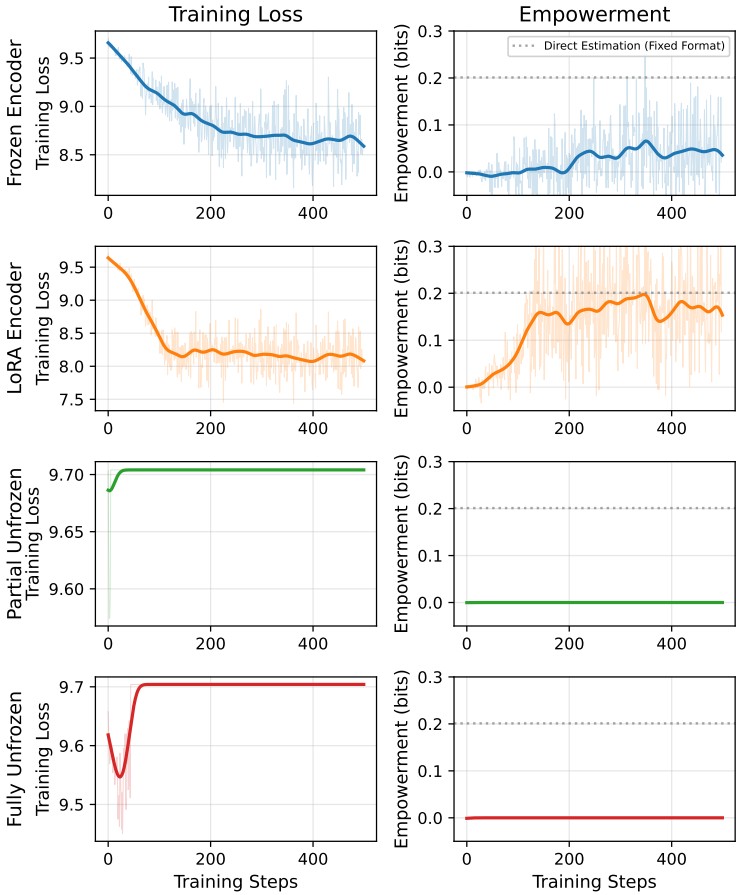

*Figure 19.* **Training curves for different fine-tuning strategies.** Comparison of training dynamics across frozen encoder, LoRA adaptation, partial fine-tuning, and full fine-tuning. LoRA maintains stable training while partial and full fine-tuning exhibit training collapse.

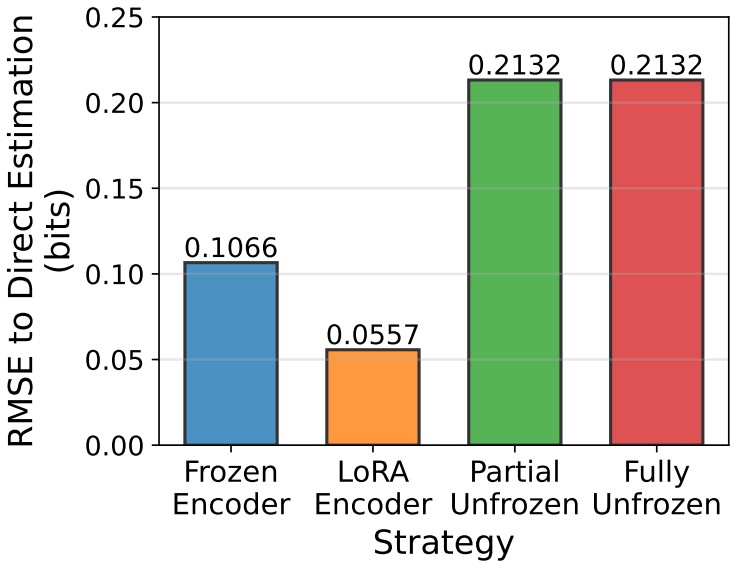

*Figure 20.* **RMSE comparison across fine-tuning strategies.** State-wise RMSE between EELMA estimates and ground-truth empowerment for each fine-tuning approach.

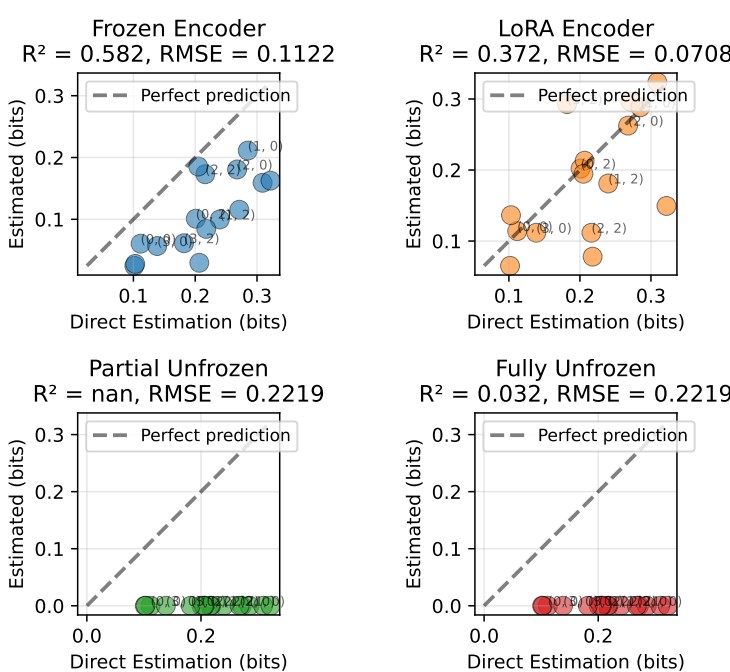

*Figure 21.* **Estimated vs. ground-truth mutual information.** Scatter plot comparing EELMA's empowerment estimates against direct computation for each fine-tuning strategy.

## M. Additional Discussion

**Applicability to Multimodal Models** Our EELMA approach is easily adaptable to multimodal language models such as vision-language models (Lin et al., 2024). The EELMA estimator can integrate embeddings of various modalities, such as

vision embeddings (Radford et al., 2021) and audio embeddings (Baevski et al., 2020), as additional inputs to the language embedding, while keeping the rest of the algorithm unchanged. We consider this a promising direction for future research.

**Power-Seeking Behavior**    Although high empowerment does not necessarily mean that the agent is power-seeking, quantifying empowerment provides a useful metric for characterizing and formalizing such behaviors without requiring explicit labels from external validators (e.g., humans). As depicted in Figure 9, empowerment-based preliminary screening via EELMA could be a valuable tool for detecting potential influential behaviors and quantifying power-seeking tendency in agent-based systems, which pose significant safety risks (Turner et al., 2021).

**Online Goal-Agnostic Evaluation.**    With the rising importance of test-time learning (Sun et al., 2025) and online preference optimization (Guo et al., 2024; Xiong et al., 2024), there is a critical need for online evaluation methods. EELMA meets this need by providing a goal-agnostic metric that approximates agent capability to track an agent's control during deployment.

**EELMA for Improving LLM Agents**    Prior work (Du et al., 2020; Eysenbach et al., 2019) has successfully used empowerment as a "goal-agnostic objective" for reinforcement learning (RL) agents in non-language environments (e.g., 2D grid worlds), but these approaches have been limited to non-language settings. Recently, (Ellis et al., 2025) applied empowerment at the token level; applying semantic state-level empowerment via EELMA as a training objective for LLM agents is a promising direction for future work.

