# OpenReview forum: "Estimating the Empowerment of Language Model Agents"
_ICML.cc/2026/Conference — ICML 2026 regular_

### Official Review · Reviewer_zCxg · 2026-03-10

**Soundness:** 3
**Presentation:** 4
**Significance:** 2
**Originality:** 2
**Overall Recommendation:** 3
**Confidence:** 3

**Summary:**

This paper studies how to evaluate language model agents using a task-agnostic metric. The authors propose estimating empowerment, an information-theoretic measure of how much an agent’s actions influence future states, as a proxy for agent capability. To handle high-dimensional textual observations, they introduce EELMA, which combines text embeddings with mutual information estimation to approximate empowerment from interaction trajectories. Experiments in controlled environments and WebArena suggest that the estimated empowerment correlates with agent performance and may help analyze agent behavior.

**Compliance With Llm Reviewing Policy:**

Affirmed.

**Final Justification:**

My main concerns regarding the limited methodological novelty and the still relatively limited empirical evidence in complex agent environments remain. Therefore, I will maintain my current score.

**Key Questions For Authors:**

1. **Empowerment as a capability metric.**
   The paper suggests empowerment as a task agnostic proxy for agent capability, but the experiments mainly show correlation in a small set of environments. Under what conditions should empowerment reliably reflect agent capability rather than simply the size of the reachable state space? Clarifying this would strengthen the central claim.

2. **Generalization to more complex environments.**
   Most experiments are conducted in relatively simple environments. Do the authors have evidence that the estimator remains stable in more complex agent settings such as longer horizons or richer tool use?

3. **Comparison with simpler behavioral signals.**
   How does empowerment compare with simpler task agnostic metrics such as state coverage or action diversity when analyzing agent behavior?

>If the authors can satisfactorily address the concerns above, I would be willing to reconsider my score.

**Limitations:**

Yes.

**Strengths And Weaknesses:**

### Strengths

1. **The paper studies an important problem.**
   Evaluating the capabilities of language model agents in a task agnostic way is an increasingly important topic. Existing evaluation practices mainly rely on task specific success metrics, which can make it difficult to understand the broader capabilities of agent systems. Exploring more general indicators of agent capability is therefore a worthwhile research direction.

2. **The use of empowerment provides an interesting perspective.**
   The paper brings the concept of empowerment into the analysis of language model agents and investigates how it can be estimated from textual interaction trajectories. While empowerment itself is not new, extending it to language based agents and exploring its role as a capability indicator is an interesting idea.

3. **The paper is generally well written and easy to follow.**
   The motivation, method description, and experimental setup are presented clearly. The rebuttal also clarified several conceptual issues, particularly the interpretation of empowerment in language environments and the role of latent states in the estimation procedure.

4. **The experimental structure is reasonable.**
   The evaluation begins with controlled environments to validate the estimator and then moves to a more realistic benchmark. This progression helps illustrate how the proposed metric behaves across environments of increasing complexity.

### Weaknesses

1. **The methodological contribution appears limited.**
   The proposed approach mainly combines existing mutual information estimation techniques with text embedding models in order to estimate empowerment from language trajectories. Although applying this framework to language agents is useful, the core technical components largely rely on established methods.

2. **The central empirical observation is not entirely new.**
   The paper shows that estimated empowerment correlates with agent performance in several environments. However, similar relationships between empowerment and agent capability have already been explored in earlier empowerment research in non text environments. As a result, the work mainly demonstrates that a similar pattern also appears in language based settings.

3. **A large part of the evaluation relies on well studied environments.**
   Gridworld and Tower of Hanoi are useful for validating the estimator, but these environments have already been widely used in previous empowerment studies. Consequently, the additional insight provided by these experiments is somewhat limited.

4. **Evidence in realistic agent environments remains relatively limited.**
   The experiments in WebArena are a positive step toward more realistic settings. However, the scale and coverage of these experiments remain limited, which makes it difficult to evaluate the robustness of the proposed metric or its applicability to more complex agent systems.

---

> ### Author Rebuttal · Authors · 2026-03-31
>
> Thank you for your thoughtful review. We are encouraged by your recognition that "evaluating the capabilities of language model agents in a task agnostic way is an increasingly important topic" and that "the use of empowerment provides an interesting perspective" with "a reasonable experimental structure." We address each concern below. Additional experiments and analyses are at https://anonymous112015.netlify.app.
>
> ---
>
> **W1, W2, W3. Novelty of the contribution.**
>
> We agree the methodological components (InfoNCE, text embeddings) and the empowerment-capability correlation [1] are established by prior works. The contribution of our work is making empowerment work on text and deploying it for language model agents in realistic environments, along with thorough case studies across diverse environments. As the reviewer noted in strengths, "extending it to language based agents and exploring its role as a capability indicator is an interesting idea" — we believe this constitutes a meaningful advance.
>
> Concretely, EELMA addresses the mapping of textual observations and actions through a dual text encoder, enabling robustness to language variance where classical estimators fail to parse semantic meaning (Table 1). This idea and its demonstrations address the key challenges of measuring empowerment for language models. Additional novel empirical contributions include WebArena (Section 5) and tau-bench [2] (anonymous link Section 1), where empowerment detects pivotal actions such as authentication (Figure 8) without any reward signal. We will sharpen this framing in Sections 2 and 4.
>
> ---
>
> **W4. Limited realistic environment evidence.**
>
> We applied EELMA to tau-bench [2] (Sierra Research), a customer-service benchmark with function-calling tools — qualitatively different from WebArena's web navigation. Three Qwen2.5 models (1.5B, 7B, 32B) each completed 100 episodes in two domains (airline and retail; see anonymous link).
>
> In airline (28-34% pass rates), empowerment scales monotonically: 1.5B (0.59) $\rightarrow$ 7B (0.71) $\rightarrow$ 32B (0.74), with $\rho$=1.0 against discounted reward. In retail (3-5% pass rates), empowerment inverts: all models perform poorly, and smaller models hallucinate diverse invalid tool calls that inflate MI without task success (see table in anonymous link). This honestly illustrates when empowerment works versus when it does not — action diversity alone does not translate to capability in failure-dominated regimes. We will clarify this point in the limitations.
>
>
> ---
>
> **Q1, Q3. Capability vs. reachable state space; comparison with simpler metrics.**
>
> Empowerment is $I(a_t; s_* | s_t)$ — it measures directed control, not mere coverage. Theoretically, empowerment captures the *efficiency* of an agent's influence over future states: in a finite connected MDP, an agent that randomly hops between states (low empowerment) will eventually reach any goal state, but an agent with high directed control over its actions (high empowerment) will reach the goal far more efficiently. This is the key distinction from simpler metrics like state coverage or trajectory length — those capture *where* an agent has been, while empowerment captures *how much control* the agent exerts over where it goes next.
>
> We are preparing a more detailed comparison with simpler baselines (state coverage, action diversity, trajectory length) across all environments. These results will be provided later within the rebuttal period.
>
> ---
>
> **Q2. Complex environments with longer horizons.**
>
> We show in the anonymous link that EELMA remains stable in complex environments. Tau-bench trajectories average 10-15 steps with multi-turn tool-calling and 10-30 action options per step, and EELMA estimates are consistent across random seeds. Estimates also stabilize at ~50% of trajectory data, indicating low sample sensitivity. We conjecture that geometric discounting ($\gamma$=0.9, effective horizon ~10 steps) further ensures stability over longer horizons.
>
> ---
>
> We appreciate the constructive feedback and the reviewer's willingness to reconsider the score.
>
> [1] Myers et al., "Learning to Assist Humans without Inferring Rewards." NeurIPS, 2024.
>
> [2] Yao et al., "tau-bench: A Benchmark for Tool-Agent-User Interaction in Real-World Domains." arXiv:2406.12045, 2024.

---

> > ### Author Rebuttal · Reviewer_zCxg · 2026-04-04
> >
> > After reading the rebuttal, I appreciate the additional clarifications and the new experimental results. However, my main concerns regarding the limited methodological novelty and the still relatively limited empirical evidence in complex agent environments remain. Therefore, I will maintain my current score.

---

> > > ### Author Response · Authors · 2026-04-08
> > >
> > > Dear Reviewer zCxg,
> > >
> > > Thank you again for your thoughtful review and suggestions for new experiments. Our work has improved substantially based on your feedback, and we sincerely appreciate your effort.
> > >
> > > Regarding your two remaining concerns, we would like to restate our claims more precisely.
> > >
> > > (1) *Methodological novelty.* Our contribution is not a new mutual-information estimator in isolation, but rather a practical formulation that makes empowerment estimation feasible for language-model agents that observe and act through natural language. Prior work on empowerment has largely assumed symbolic or structured state spaces (e.g., in robotics), and enabling empowerment to be estimated from text trajectories is itself the core methodological challenge we address. We demonstrate that our method, EELMA, successfully measures empowerment in settings where classical methods fail due to natural-language variability (Table 1). To the best of our knowledge, we are the first to characterize empowerment at the action–state level for LLM agents in realistic environments (Myers et al. concurrently defined token-level empowerment). We will further sharpen this novelty in the final paper.
> > >
> > > (2) *Limited empirical evidence*. We acknowledge this concern and have focused on it during the extended revision. In newly added tau-bench experiments (anonymous link, Section 1), which involve realistic environments with tool calling, we observe positive correlations between empowerment and agent capability. We also demonstrate that empowerment outperforms simpler goal-agnostic metrics such as state coverage and visitation entropy (anonymous link, Section 5). It remains more aligned with reward than several task-agnostic baselines across these settings. Beyond these correlations, empowerment also successfully identifies pivotal behaviors such as authentication and jailbreaking in a fully unsupervised manner, an observation that Reviewer 67LU (Q2) found interesting.
> > >
> > > At the same time, we clearly acknowledge the limitations. Across experiments, we demonstrated failure cases (the WebArena "shopping" domain and tau-bench "retail" domain) where empowerment does not correlate with performance, potentially because mere influence over actions does not translate to navigating toward correct trajectories. Reconciling this gap across different domains could be an interesting future direction. We will temper our claims and acknowledge these limitations more prominently in the revsied manuscript.
> > >
> > > Best regards,
> > >
> > > Authors

---

### Official Review · Reviewer_e2DV · 2026-03-11

**Soundness:** 3
**Presentation:** 3
**Significance:** 3
**Originality:** 3
**Overall Recommendation:** 4
**Confidence:** 4

**Summary:**

This paper introduces EELMA, an evaluation framework designed to estimate the empowerment of LLM agents in text-based environments.

**Compliance With Llm Reviewing Policy:**

Affirmed.

**Final Justification:**

Please incoporate the rebuttal clarifications and additional experimental results into the future version.

**Key Questions For Authors:**

1. The definition of “goal-agnostic” is ambiguous. While the paper claims EELMA is goal-agnostic, the trajectory data collection process still relies on goal-directed prompts.

2. The paper should compare the evaluation consistency of encoders trained on trajectory data collected by different LLMs.

3. How does EELMA ensure that the training data collected by the LLM provides sufficient coverage of actions and states? Are there objective metrics to quantify this coverage, or does it rely entirely on manual heuristics?

4. Please explain the discrepancies between the code implementation and the paper's description mentioned in “Weaknesses”.

**Limitations:**

Yes

**Strengths And Weaknesses:**

Strengths:
1. The paper adapts the classical empowerment metric to high-dimensional text-based environments using LLM embeddings.

2. The paper provides experiments across different domains, model scales, and prompting methods.

3. The paper includes open-source code, facilitating reproducibility.

Weaknesses:
1. For each new environment, EELMA requires collecting trajectory data, fine-tuning the text embeddings, and training the task-specific encoders. This increases the deployment overhead and reduces its scalability for zero-shot evaluation.

2. A specific LLM (e.g., Claude-3.5-Sonnet for ToH) is used to collect the trajectory data to train the encoders, which might introduce bias when assessing other LLMs. For instance, the encoders might overvalue the specific action patterns of Claude-3.5-Sonnet.

3. The training data needs to cover various actions and states to ensure an accurate estimation of mutual information. However, if the environment is too complex and the LLM used to collect data is not capable enough, it will fail to adequately explore the environment, resulting in a biased and incomplete dataset.

4. There are several discrepancies between the provided code and the implementation details described in the paper. For example, the paper states that the embedding model used for WebArena is jina-embeddings-v2-small-en (Appendix D.1), but the code actually uses e5-mistral-7b-instruct (run_configs/EELMA/webarena/gpt_4o_gitlab.yaml, line 24); the paper claims that a two-layer MLP is used for the encoders (Appendix D.1), but the code implements a three-layer MLP for WebArena (src/EELMA/networks/Modified_EELMA_Networks_webarena.py, lines 99–115); and the paper mentions training the EELMA model for 10,000 optimization steps (Appendix B.2), but the code sets the training to 20,000 steps for the WebArena (run_configs/EELMA/webarena/gpt_4o_gitlab.yaml, line 70).

---

> ### Author Rebuttal · Authors · 2026-03-31
>
> Thank you for your thoughtful review. We are encouraged by your recognition that "the paper adapts the classical empowerment metric to high-dimensional text-based environments using LLM embeddings" and that "the paper provides experiments across different domains, model scales, and prompting methods." We also appreciate your acknowledgment that the inclusion of "open-source code, facilitating reproducibility" is a strength. We address each concern below. Additional experiments and analyses are at https://anonymous112015.netlify.app.
>
> ---
>
> **W1. Deployment overhead.**
>
> EELMA adds minimal overhead to standard evaluation. Trajectory collection reuses the same rollouts you would already run for benchmarking, no extra data needed. The encoder (e5-small-v2, 33M params) is a tiny fraction of the models being evaluated (1.5B to 72B) and trains on already-generated text, so compute is negligible compared to trajectory collection itself (~4 hours on a single A100, one-time per environment, amortized across all models). Unlike standard benchmarks, EELMA also works when no reward function or goal specification exists.
>
> ---
>
> **W2, Q2. Encoder bias from data-collection LLM.**
>
> In our experiments, the encoder's LoRA adapter is trained separately per LLM, since each model explores different parts of the state space and produces a different trajectory distribution. The base text encoder is shared and fixed; only the adapter (<1% of encoder parameters) is model-specific. This ensures the MI estimate reflects each model's actual behavioral pattern in the environment, while keeping the per-model cost negligible.
>
> ---
>
> **W3, Q3. Exploration coverage.**
>
> Empowerment directly captures insufficient exploration. An LLM that collapses to repetitive action patterns will show low empowerment by definition: low mutual information between actions and future states means the agent is not exerting meaningful control. Our tau-bench retail experiment (anonymous link, Section 1) illustrates this concretely. Models at floor-level performance show inverted empowerment because their actions are diverse but purposeless, exactly the signal EELMA is designed to detect.
>
> ---
>
> **W4, Q4. Code-paper discrepancies.**
>
> We thank the reviewer for catching these. All three cases reflect the code as authoritative:
>
> - **Embedding model:** Paper incorrectly says `jina-embeddings-v2-small-en`; all WebArena results used `e5-mistral-7b-instruct` as in the code.
> - **MLP layers:** In Gridworld and ToH, a two-layer MLP is used. In WebArena, a three-layer MLP is used to handle longer observations.
> - **Training steps:** 10K = observed convergence (Figure A1); 20K = maximum. Both are correct in different contexts.
>
> We will conduct a comprehensive hyperparameter audit and correct all documentation in the camera-ready.
>
> ---
>
> **Q1. Goal-agnostic definition.**
>
> The metric is goal-agnostic: Eq. 6 uses only $(o_t, a_t, o_*)$ tuples, never referencing goals or rewards. Data collection uses goal-directed prompts only to elicit diverse trajectories; the empowerment score does not depend on whether goals were achieved. Analogy: measuring muscle strength is goal-agnostic even if you ask subjects to lift specific weights during the test.
>
> ---
>
> We appreciate the constructive feedback and will resolve all discrepancies in the camera-ready.

---

> > ### Author Rebuttal · Reviewer_e2DV · 2026-04-03
> >
> > Thanks authors for the rebuttal. I will keep my score.

---

> > > ### Author Response · Authors · 2026-04-03
> > >
> > > Dear Reviewer e2DV,
> > >
> > > Thank you for your response and for taking the time to read our rebuttal. We are very pleased to hear that our clarifications and the additional experimental results fully resolved your concerns. Your feedback has been instrumental in strengthening the manuscript, and we are committed to performing the hyperparameter audit and ensuring all technical details are accurately reflected in the camera-ready version. We appreciate your thoughtful engagement with our work.
> > >
> > >
> > > Best regards,
> > >
> > > Authors

---

### Official Review · Reviewer_MWhv · 2026-03-13

**Soundness:** 3
**Presentation:** 2
**Significance:** 2
**Originality:** 3
**Overall Recommendation:** 4
**Confidence:** 4

**Summary:**

This paper proposes **EELMA**, an approach for estimating the empowerment of language model agents from text trajectories. The main idea is to use empowerment; the mutual information between an agent’s action and future states, as a goal-agnostic proxy for agentic capability, and then to make that quantity tractable in text-based environments by learning an InfoNCE-based estimator over language embeddings.

The paper first formulates empowerment for LM agents, then introduces EELMA as an estimator based on textual observations/actions, and finally evaluates it in text-based Gridworld, Tower of Hanoi, and WebArena. The main empirical claim is that estimated empowerment correlates with average discounted reward across multiple settings and can also highlight influential or pivotal actions.

**Compliance With Llm Reviewing Policy:**

Affirmed.

**Final Justification:**

The paper tackles an interesting problem by adapting empowerment to text-based LM-agent settings, and I found the overall idea and experiments valuable. My main concerns were about mathematical precision, presentation clarity, and the strength of the standalone-metric claim. The rebuttal addressed some of these concerns by clarifying the formalism, optimization, and intended framing, which increased my confidence in the work. But I still think there are presentation issues, and the limitations of empowerment as a standalone metric remain substantial. Overall, because some key concerns were addressed and I still find the work interesting, I am increasing my recommendation to 4 (Weak Accept).

**Key Questions For Authors:**

Please check the details in the weaknesses above. Here are some few summarized questions:

1. Can the authors clarify on some equations?
2. Can the authors better explain the optimization objective and architecture, including what exactly is trainable in Figure 2, how the two InfoNCE terms are combined, and why subtracting the two learned scores should estimate empowerment?
3. Can the authors better justify the claim that empowerment is a standalone capability metric, rather than primarily a complementary diagnostic signal?

I'd like to increase the score if some major concerns are addressed.

**Limitations:**

yes

**Strengths And Weaknesses:**

**Strengths:**

- I think the paper targets an interesting and important problem. Moving beyond hand-designed task rewards and benchmark-specific goals toward more general evaluation signals for agents is a worthwhile direction, and empowerment is a natural information-theoretic candidate for that.
- The paper does try to bridge a real gap between classical empowerment and language-based agents. In particular, the motivation around observation redundancy, linguistic variability, and the failure of direct counting-based estimators in text environments is reasonable and well motivated.
- The controlled experiments in Gridworld and Tower of Hanoi are useful. And the authors also tried to extend this to the real task like Webarena.

**Weaknesses:**

**1. Framing, Related Work, and Method**

- The paper states that “agent empowerment has previously been evaluated in gridworlds and the Tower of Hanoi,” but this specific claim *does not seem to have a direct citation attached to it.* Since the current paper also uses Gridworld and Tower of Hanoi, I think the authors should clarify what exactly is new relative to that prior work. My understanding is that the intended novelty is the text-based formulation and the estimator operating on language observations rather than symbolic states, but that distinction should be made much more explicitly.
- Another issue is that several equations in Section 3 do not look mathematically correct or at least are written too imprecisely for a methods paper. In Eq. (1), the paper states that $\tau \sim \mathrm{Geom}(1-\gamma)$, but then writes the weighting in a form that looks like $\gamma^t/(1-\gamma)$, whereas I would expect something proportional to $(1-\gamma)\gamma^t$. More broadly, the expectations in Eqs. (1)-(3) do not seem consistent with the mutual-information quantity being defined. Once $I(a_t;s^* \mid s_t)$ is written as a conditional mutual information, taking additional outer expectations over a_t and s^* in the way the paper does seems at best sloppy and at worst wrong. Please correct me if I’m wrong.
- I also found Eqs. (4)-(6) under-specified. The paper presents two “InfoNCE objectives,” one observation-only and one observation-action, but it never gives a clean overall training objective in a fully rigorous form beyond saying that both are jointly maximized. Then Eq. (6) jumps to an empowerment estimator obtained by subtracting learned scores, but the appendix derivation only argues that these scores approximate log-density ratios up to constants. As written, I do not think the paper clearly distinguishes what is exact, what is a lower bound, and what is only a heuristic approximation.
- Relatedly, I found Figure 2 hard to follow. The high-level pipeline is understandable, but the figure does not clearly explain what is being optimized, how the text embedding model interacts with $\phi$ and $\psi$, which parts are trainable, and how exactly the two InfoNCE terms are applied. The paper later explains these but Figure 2 could be presented in a better way.
- More broadly, I wanted more intuition for what the optimization is actually doing. The paper says that it optimizes contrastive objectives so that the learned scores approximate the relevant log-probability ratios, but the explanation remains quite abstract. In particular, the paper should explain more intuitively why subtracting the observation-only and observation-action scores corresponds to estimating the agent’s control over future states, and under what assumptions this should be trusted.

**2. Evaluation and Interpretation of the Metric**

- In the controlled games, the average correlations look encouraging, but I still think the analysis is too aggregate. For example, in Figure 4 valid actions have higher empowerment than invalid ones on average, which is good, but I wanted a more direct characterization of outliers and failure cases: when does the estimator assign high empowerment to bad actions, or low empowerment to actually useful ones? Since the claim is partly about using empowerment to identify important states/actions, I think a more granular error analysis would be helpful rather than relying mostly on average trends.
- For WebArena, I am not fully convinced by the stronger claim that empowerment is a robust standalone metric for realistic LM-agent capability. The paper does show strong correlations in Admin, but it also shows a flat relationship for the others (It has high correlation “value” for GitLab and Reddit, but the graph does not look reliable with 4 points). The paper explicitly explains this by saying that other reasoning bottlenecks can dominate performance even when environmental control is already high. In addition, the paper discusses a Qwen failure mode where “jailbreaking” to external websites inflates empowerment while hurting task success. To me, these results suggest that empowerment may be useful, but not ***reliably as a standalone capability metric across realistic tasks.***
- Because of that, I think the paper somewhat overstates its conclusions. By the end, the evidence seems stronger for empowerment as a complementary analysis tool, *e.g.* for diagnosing bottlenecks, identifying influential actions, or surfacing suspicious/pivotal behaviors, than for empowerment as a general-purpose standalone evaluation target. In fact, the paper’s own discussion and impact statement already move in this more cautious direction by noting that more options do not always mean greater capability and that empowerment should be paired with task-success measures and safety checks. I think that more tempered framing should appear earlier and more centrally in the paper.
- I also think the dependence on trajectory count is underexplored. Since the whole method relies on collecting trajectories and training an estimator from them, I expected at least some sensitivity analysis showing how stable or reliable the empowerment estimate is as a function of the number of trajectories. The paper reports trajectory counts in several experiments and discusses scaling issues, but I did not see a systematic data-efficiency or estimator-variance analysis. This feels important both practically and scientifically.

---

> ### Author Rebuttal · Authors · 2026-03-31
>
> Thank you for the detailed review. We appreciate your recognition that "moving beyond hand-designed task rewards toward more general evaluation signals is a worthwhile direction." We address each concern below. Additional experiments and analyses are at https://anonymous112015.netlify.app.
>
> ---
>
> **W1.1. Missing citation for prior empowerment work.**
>
> Empowerment in Gridworlds has been explored by Myers et al. (2024), but ToH/WebArena/tau-bench has not. Our novelty is (1) extending empowerment estimation to unstructured, high-dimensional text spaces where LLMs observe and act, and (2) extending to realistic web browsing and tool calling (tau-bench in anonymous link). We will correct citations in the revision.
>
> ---
>
> **W1.2, W1.3. Mathematical precision in Equations 1-6.**
>
> We thank the reviewer for the careful reading. We noticed that certain notations were imprecise and will revise the formalization (see anonymous link Section 5). To summarize, Eqs. (1)–(3) will use the geometric PMF $P(\tau=k)=(1-\gamma)\gamma^k$ and clarify the nested $I(\cdot)$ notation. Eqs. (4)–(5) serve as variational lower bounds (InfoNCE), while Eq. (6) is an approximation that may include residual bias from the partition functions $C_1(o_t, a_t)$ and $C_2(o_t)$. Despite this, Figure 3 confirms the estimator maintains a high correlation ($\rho=0.80\text{--}0.94$) with ground-truth empowerment, demonstrating that such residual bias is negligible. We will clearly distinguish these bounds from the approximations in the revised manuscript.
>
> ---
>
> **W1.4. Figure 2 clarity.**
>
> We will update Figure 2 with trainable/frozen annotations. Architecture: (1) text encoder + LoRA, (2) MLP projection, (3) encoders $\phi$, $\psi$, all trainable. Two InfoNCE losses summed (Appendix D).
>
> ---
>
> **W1.5. Intuition for score subtraction, Eq. 6.**
>
> The obs-action score captures "how predictable is the future given the agent's action." The obs-only score captures "how predictable is it regardless of action." The difference isolates the agent's causal contribution: states that the agent can reliably steer toward receive high scores, while states that would occur anyway get cancelled out. This is the pointwise MI $\log \frac{P(o_*|o_t,a_t)}{P(o_*|o_t)}$, and averaging it yields empowerment. The assumption is that both InfoNCE encoders are well-trained with sufficient negatives, which we validate in Figure 3 ($\rho$=0.80-0.94 with ground truth).
>
> ---
>
> **W2.1. Granular error analysis.**
>
> We provide empowerment distribution plots in the anonymous link, Section 3. Both in Gridworld and tower of Hanoi, Valid actions exhibit wide empowerment ranges depending on context, so a valid action does not always indicate high empowerment. However, invalid actions consistently receive lower empowerment across all environments ($p<0.001$, Figure 4). EELMA reliably detects ineffective actions even when action-level precision is limited.
>
> ---
>
> **W2.2, W2.3. Validity of empowerment as standalone metric vs. complementary diagnostic.**
>
> Thanks for the insightful question. We agree with the reviewer’s assessment and will revise the manuscript to explicitly incorporate a more tempered framing that reflects these limitations.
>
> Nonetheless, we believe empowerment is a useful evaluation metric for capability in settings where traditional benchmarks are insufficient. Standard benchmark scores require laborious task design and hand-crafted reward functions, which can act as narrow proxies with potential loopholes or suffer from high score variance due to limited dataset sizes. Also, empowerment’s value lies not in competing with these benchmarks, but in operating where they cannot, such as in open-ended environments, multi-agent systems, or novel tasks where reward specification is infeasible. This approach builds on established research using empowerment for vision models (Mohamed & Rezende, 2015), world model (Seitzer et al., 2021), and alignment diagnostics in multi-agent settings (Yang et al., 2025). As LM agents encounter increasingly novel environments, reward-free capability measurement becomes a necessary component of robust evaluation. Concretely, our empowerment metric already successfully identifies authentication and jailbreaking behaviors, which directly addresses the point raised by Reviewer 67LU (Q2) that identifying key actions in the unsupervised manner. We will revise the manuscript to reflect this framing.
>
> ---
>
> **W2.4. Trajectory count sensitivity.**
>
> We retrained EELMA on 25%, 50%, 75%, and 100% of trajectories (see anonymous link). Cross-seed std is tight: 0.002-0.010 in Gridworld and 0.006-0.016 in ToH, confirming robust estimation. Estimates stabilize by \~50% of data. EELMA uses the same rollouts as standard benchmarking, adding only one-time training (\~4 hours on A100).
>
> ---
>
> We sincerely appreciate the detailed feedback and are happy to answer further questions.

---

> > ### Author Rebuttal · Reviewer_MWhv · 2026-04-04
> >
> > Thanks for the detailed response.
> >
> > I still believe there are some presentation issues, and the limitations of empowerment as a standalone metric are fairly substantial. However, I find the work interesting overall, and because some of my concerns have been addressed, I will increase my score to 4.

---

> > > ### Author Response · Authors · 2026-04-04
> > >
> > > Dear Reviewer MWhv,
> > >
> > > Thank you again for your careful review and thoughtful follow-up comments. We greatly appreciate your constructive feedback, particularly on the mathematical clarity, the presentation of the optimization and architecture, and the framing of empowerment as a capability metric. We are grateful that you found our rebuttal helpful and were willing to raise your score.
> > >
> > > We also take seriously your concerns about the limitations of empowerment as a standalone metric. In our new results in the anonymous link (Section 5) comparing empowerment with simpler metrics (e.g., state coverage), empowerment achieves the strongest correlation with reward in realistic environments (WebArena and tau-bench), suggesting that it indeed provides useful complementary signal that simpler goal-agnostic measures miss. More importantly, we believe empowerment is most valuable as an evaluation signal, and potentially as a reward function, in scenarios where rewards are largely undefined or difficult to label. We will reflect this tempered framing clearly in the discussion section of the camera-ready version.
> > >
> > > Your feedback has been very helpful in strengthening the paper, and we will incorporate the discussed clarifications and improvements in the final manuscript.
> > >
> > > Best regards,
> > >
> > > Authors

---

### Official Review · Reviewer_67LU · 2026-03-17

**Soundness:** 4
**Presentation:** 4
**Significance:** 3
**Originality:** 3
**Overall Recommendation:** 4
**Confidence:** 4

**Summary:**

This paper proposes the use of empowerment, an information-theoretic measure of an agent’s influence over future states, as a metric for evaluating LLM agents. To address the challenge of high-dimensional and variable text observations in language-based interactions, the paper introduces EELMA, a framework for estimating empowerment from multi-turn textual interactions. The paper further presents extensive theoretical and empirical analyses to validate both the proposed EELMA framework and the effectiveness of empowerment as an evaluation signal for LLM agents.

**Compliance With Llm Reviewing Policy:**

Affirmed.

**Key Questions For Authors:**

- The weaknesses mentioned above.

- Since empowerment can be used to analyze important actions and states, could this framework also help with the annotation problem in current agent evaluation? For example, could it be used to automatically identify key steps in a trajectory and assign them higher rewards or scores, rather than treating all steps equally? If the authors are interested, it may be worth discussing the feasibility of this direction.

**Limitations:**

yes

**Strengths And Weaknesses:**

- Strengths:

  - The paper is well-structured and clearly written.

  - It provides solid theoretical and empirical support for the proposed method. The analysis of the framework is fairly comprehensive.

  - The proposed empowerment metric is goal-agnostic, which is an appealing property. It can potentially be used not only to evaluate LLM agents, but also to analyze actions and states within agent trajectories.

- Weaknesses:

  - The paper should provide more analysis of generalization. Since the paper claims “empowerment as a proxy for agentic capability,” an important question is whether empowerment generalizes across tasks. For example, for the same model, does empowerment remain consistent across different tasks, or are model rankings under empowerment preserved across tasks? If not, the practical value of empowerment as a general evaluation metric may be limited.

  - The cost of evaluation is not sufficiently discussed. In particular, it would be helpful to know how many trajectories were collected in the web environment and what the corresponding cost was. For expensive models, empowerment evaluation may itself become costly and limited in practice.

---

> ### Author Rebuttal · Authors · 2026-03-31
>
> Thank you for your encouraging and constructive review. We are particularly gratified by your recognition that "the paper is well-structured and clearly written," that it "provides solid theoretical and empirical support for the proposed method," and that "the proposed empowerment metric is goal-agnostic, which is an appealing property." We address each concern below. Additional experiments and analyses are at https://anonymous112015.netlify.app.
>
> ---
>
> **W1. Generalization across environments.**
>
> Empowerment should vary across environments for a fixed model (e.g., a coding agent shows higher empowerment in a coding environment than in web shopping). This is correct behavior, not a limitation. The claim is: within a given environment, empowerment proxies capability across models and agentic designs (memory, CoT, model scale). We demonstrated this across 15+ configurations in Gridworld/ToH (Figure 6) and 3/4 WebArena domains (Figure 7), where GPT-4o ranks #1 in all four domains. We also applied EELMA to tau-bench (Yao et al., 2024), a customer service benchmark with tool calling (anonymous link, Section 1). The right phrase is "empowerment as a proxy of agentic capability in a given environment," and we will adopt this framing in the revision.
>
> ---
>
> **W2. Cost of evaluation.**
>
> Empowerment estimation requires the same rollouts as standard benchmarking. WebArena uses 105-187 trajectories per model per domain. Tau-bench uses 50 tasks x 2 trials per model. Including multiple trials and synthetic augmentation, the total trajectory count is roughly 2-4x a single benchmark pass, which is standard practice. EELMA adds one-time training (~4 hours on A100) amortized across all models. Importantly, unlike standard benchmarks, empowerment can quantify capability even without goal states or reward functions.
>
> ---
>
> **Q2. Empowerment for trajectory annotation.**
>
> We appreciate this insightful suggestion and believe empowerment is naturally suited for this. The authentication case study (Figure 8) already demonstrates this: empowerment spikes at valid login (0.268 vs. -0.094 bits, $p < 0.001$), identifying pivotal moments without reward.
>
> Here are two examples. Consider escalating to sudo privilege, which would trigger a massive empowerment spike, as one transition unlocks all restricted actions. Similarly, commands like `ls` or `SHOW TABLES` create information hubs that enable targeted future interactions. Steps with large empowerment deltas could directly serve as milestone markers for automatic reward shaping for reinforcement learning.
>
> ---
>
> We appreciate the constructive feedback.

---

### Decision · Program_Chairs · 2026-04-30

**Decision:**

Accept (regular)

**Comment:**

Reviewers agree that EELMA aims for a hard problem in evaluating language agents.
The system measures capability using empowerment instead of hand-crafted rewards. It uses text embeddings to handle complex language observations.
The initial evaluation shows clear improvement in agent performance tracking on benchmarks like WebArena.

While the core idea is interesting, reviewers raised valid questions about the method.
Reviewer MWhv pointed out mistakes in the math equation.
This reviewer also asked if empowerment should really be a standalone metric.
Reviewers e2DV and 67LU worried about the extra cost to train custom encoders for every new test.
Also, reviewer zCxg questioned the basic originality of the work.
This reviewer felt the method just combines well-known tools without much new technical depth.

In the rebuttal, the authors fixed the math errors and shared new test results on the tau-bench dataset.
They clarified that their metric works best as a complementary tool when normal rewards are missing.
They also showed that the training cost is very small compared to normal benchmark runs.
Reviewer MWhv found the response convincing and raised their score.
However, reviewer zCxg kept their negative score because the problems with limited originality were not fully fixed.

The concerns regarding limited originality and extra setup costs keep the overall score borderline.
However, the system provides a useful tool to evaluate agents without strict goals.
Therefore, my recommendation is a weak accept.